# Looking Beyond the Top-1: Transformers Determine Top Tokens in Order

## Abstract

Understanding the inner workings of Transformers is crucial for achieving more accurate and efficient predictions. In this work, we analyze the computation performed by Transformers in the layers after the top-1 prediction has become fixed, which has been previously referred to as the "saturation event". We expand the concept of saturation events for top-$k$ tokens, demonstrating that similar saturation events occur across language, vision, and speech models. We find that these saturation events happen *in order* of the corresponding tokens' ranking, i.e., the model first decides on the top ranking token, then the second highest ranking token, and so on. This phenomenon seems intrinsic to the Transformer architecture, occurring across different architectural variants (decoder-only, encoder-only, and to a lesser extent full-Transformer), and even in untrained Transformers. We propose an underlying mechanism of task transition for this sequential saturation, where task $k$ corresponds to predicting the $k$-th most probable token, and the saturation events are in fact discrete transitions between the tasks. In support of this we show that it is possible to predict the current task from hidden layer embedding. Furthermore, using an intervention method we demonstrate that we can *cause* the model to switch from one task to the next. Finally, leveraging our findings, we introduce a novel token-level early-exit strategy, which surpasses existing methods in balancing performance and efficiency.

## 1 Introduction

In recent years, Transformer-based models (Vaswani et al., 2017) have achieved state-of-the-art performance in various tasks across multiple modalities, including text generation, image classification, and automatic speech recognition (Zhang et al., 2023; OpenAI et al., 2024). This has lead to a growing interest in model interpretability, which tries to explain the internal processes that give rise to these remarkable capabilities. In the language domain, investigation into the way the model's predictions are constructed has led to the discovery of *saturation events*, where the model's top-1 prediction is determined in an early layer and remains fixed in subsequent layers (Geva et al., 2022).

In this work, we address the following question – *what computation is the Transformer model performing after the saturation event?* Taking inspiration from Frydenlund et al. (2022), we treat the model's output as a ranking over the labels instead of a probability distribution. Using the logit lens (Nostalgebraist, 2020), we project the hidden states of intermediate layers onto the vocabulary space to extract ranking over tokens and analyze the changes over the layers. For the first time, we show that in a decoder-only text Transformer (GPT2-XL; Radford et al., 2019) saturation events also take place for the top ranking tokens beyond the top-1 (2nd, 3rd, 4th, etc.). Surprisingly, we find that they happen *in order* of their ranking, i.e. the second-ranking token is determined only after the first-ranking token, and so forth (see in Figure 1). We then generalize the results across different modalities and Transformer variants, including pretrained Transformers for both vision (encoder-only ViT-L/16; Dosovitskiy et al., 2020) and speech (encoder-decoder Whisper-large; Radford et al., 2023). Next, we demonstrate that sequential saturation seems intrinsic to the Transformer architecture, occurring even in an *untrained* randomly initialized model (GPT2-XL).

We propose that this phenomenon is due to a discrete *task-transition* mechanism, where each task $i$ corresponds to the model determining the $i$-th token in the final ranking, and the transition between one task and the next happens at the corresponding saturation layer. Furthermore, we find that the

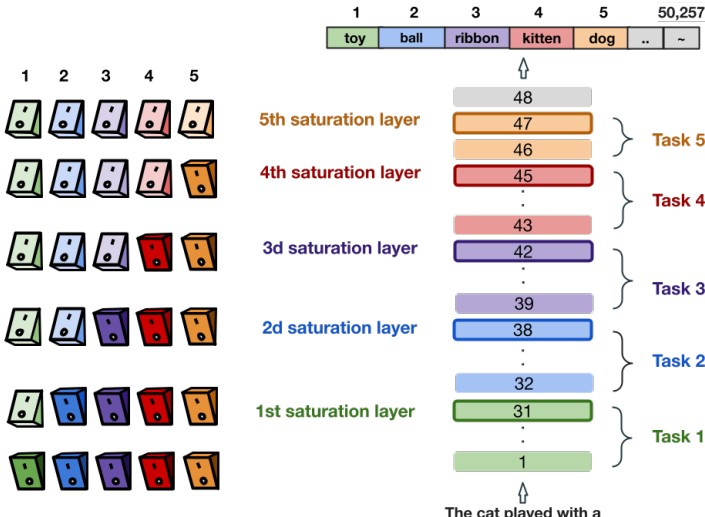

Figure 1: An illustration of the proposed task-transition mechanism wherein the layers of the Transformer perform a changing number of tasks in order, so that task $i$ is determining the $i$-th token in the final ranking, and the transition between task $i$ and task $i + 1$ occurs at the corresponding $i$-th saturation layer. The transition is akin to a switch being flipped "on" and staying "on" for the remaining layers representing the $i$-th token being fixed from this point onward.

task information is encoded in the layer embeddings and that at each saturation layer, a discrete "switch" is flipped, signaling that the relevant token has been determined, causing the model to move on to the next task while keeping this token fixed in subsequent layers. To support this, we show that it is possible to predict the task index from the layer embeddings using a simple logistic regression classifier, and that we can cause the model to transition from the first to the second task by "injecting" embeddings from either the top-1 saturation layer or of one of the subsequent layers.

Finally, we show that our findings lend themselves to practical applications in improving both model efficiency and accuracy. Based on this new understanding of the Transformer, we define a new early-exit decision strategy for text generation. Early exiting is a technique where the model can make a prediction and terminate the computation before reaching the final layer, thus improving efficiency (Schwartz et al., 2020). In our method, the early-exit layer is the first one predicted to belong to task 2, presuming that after the transition from task 1 to task 2, the top-1 token represents the model's final prediction. We show that this strategy outperforms existing token-level measures, such as softmax-response and hidden-state saturation (Schuster et al., 2022). In addition, we show that we can use task information to achieve more accurate language models, by demonstrating that in cases where the top-1 prediction is incorrect, the second highest ranking token represents a much more accurate prediction when it reaches saturation than when it does not.

Our main contributions are:

- We find that Transformers tend to decide their top ranking tokens in order, so that the top ranking token is fixed first, then the second-ranking token at a later layer and so on. We show that this occurs across various modalities and variants of the Transformer architecture, and even in untrained randomly initialized models.

- We show that sequential saturation can be explained with a discrete *task-transition* mechanism, encoded in the representation of hidden layers where each task corresponds to determining the next ranking token. We empirically show that it is possible to predict the task index only from internal activations, and that we can cause the model to switch from one task to the next via an intervention procedure.

- We show that these observations can be leveraged to achieve better downstream performance in early exiting and language modeling, in terms of both accuracy and efficiency.

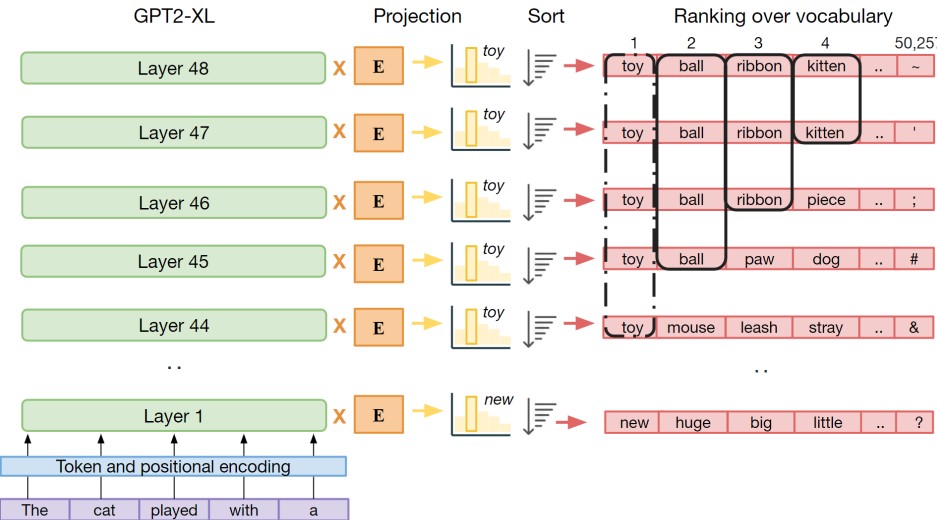

Figure 2: Schematic of our framework and visualization of the ordered saturation of the top-k tokens on GPT2-XL. The hidden states from each layer are projected onto the vocabulary space using the unembedding matrix $E$, then sorted in descending order and treated as rankings. The saturation effect is marked separately for each token in the top-4 of the final ranking, emphasizing the fact that the 2nd token saturates *after* the 1st token, the 3d token saturates *after* the 2nd token and so on. The dashed line represents the previously established saturation event of the top-1 token.

We will make the code for all of our experiments and evaluations publicly available.

## 2 EXPERIMENTS

In this section, we formulate two experiments to understand what computation the Transformer is performing in the layers after the top-1 saturation event. To achieve this, we first extend the formal definition of top-1 saturation to account for arbitrary i-th ranking token saturation (Section 2.1). Then, in Section 2.2, we leverage this definition to develop a metric capturing the extent to which the top tokens are saturated in order. In Section 2.3, we describe a probing method measuring whether it is possible to determine the rank of the token currently being considered by the model solely from the intermediate representation of the layer, without any additional context.

### 2.1 DEFINING SATURATION LAYERS

**Definition 2.1** (1st Saturation Layer; Geva et al., 2022). The saturation event occurs at layer $l$ (from here on referred to as the "1st saturation layer") for index $i$ in the input if the top-1 prediction of the model remains constant in all subsequent layers after $l$. Formally, given a model with $N$ layers, a saturation event occurs at layer $l \leq N - 1$ if for all layers $l'$ s.t. $l < l' \leq N$ the top-1 token in the ranking induced by that layer remains unchanged. For example, the saturation event marked with a dashed line in Figure 2 occurs on layer 44, since in subsequent layers the top predicted token ("toy") remains constant.

**Definition 2.2** ($k$-th Saturation Layer). Here, we are interested in examining model behavior beyond the determination of the top ranking token and so naturally extend the definition of top-1 Saturation (Definition 2.1) to capture the layer at which the $k$-th top token is determined and remains fixed. Formally, the saturation event for the $k$-th top token at index $i$ in the input occurs at layer $l^k \leq N - 1$ (from here on referred to as the " k-th saturation layer") if for all following layers $l'$ s.t. $l^k < l' \leq N$ the token in position $k$ in the ranking induced by that layer remains unchanged. We note that the saturation event defined in (Geva et al., 2022) happens at $l^1$. For example, in Figure 2, $l^1 = 44$ as that is where the top token ("toy") is determined; $l^2 = 45$, since the second-most probable token ("ball")

is determined at layer 45; and similarly $l^3 = 46$, since the third-most probable token ("ribbon") is determined at layer 46, etc.[1] See Appendix B.1 for statistics of 1st and k-th saturation.

## 2.2 Examining the Order of Saturation Layers

We investigate whether the saturation layers of the top-k tokens $l^1, l^2, \ldots l^k$ are arranged in order, i.e., whether the saturation of the first token happens before the saturation of the second token, the saturation of the second token happens before the saturation of the third token and so on. To this end, for each token in the input we first calculate these saturation layers for $k = 1, \ldots, 5$ according to Definition 2.2 and then for each $k$ compute the rank of the saturation layer of the $k$-th top token. We use $k = 5$ to ensure consistency across models, as it is the highest value of $k$ where the $k$-th token reaches saturation in at least $5\%$ of input tokens for all the models analyzed. Following our example from Figure 2, we have $l^1 = 44, l^2 = 45, l^3 = 46, l^4 = 47,$[2] and their ranking is $[1, 2, 3, 4]$, since $l^1 < l^2 < l^3 < l^4$. If the tokens reach saturation in order of ranking, as they do in this case, we would expect the average rank of the saturation layers to increase monotonically with $k$.

## 2.3 Probing for Task Transition

We argue that the mechanism underlying the saturation of the top-k tokens in order is one of task transition, such that determining the identity of each token in the final ranking is a separate task, and the model performs them sequentially: first determining the identity of 1st token, then the identity of the 2nd token, and so on, and that the transition from one task to the next occurs at the corresponding saturation layer. Additionally, we claim that the specific task number can be inferred from the model embedding at each layer, and that this information is independent of the context or the specific token predicted by the model.

To test this hypothesis, we perform a type of probing in which we train a simple one-versus-all multi-class logistic regression classifier to predict the number of the task the model is "working on" from the hidden state embeddings of the model. We collect the data for training by extracting the model's hidden states during inference and categorize them into 5 classes according to the saturation layers of the top-5 tokens for each instance. This means that for a given input, embeddings from layers up to (and including) the 1st saturation layer are classified as belonging to task 1, embeddings from layers from the next layer until the 2nd saturation layer are classified as belonging to task 2, and so forth, for tasks 1 through 5. For example, in the case of the token "a" as depicted in Figure 2, the embeddings from layers 1 through 44 would be classified as belonging to task 1; the embedding of layer 45 would be classified as belonging to task 2; the embedding of layer 46 would be classified as belonging to task 3; the embedding of layer 47 would be classified as belonging to task 4; and as the model reaches the last layer directly afterward there would be no embedding classified as belonging to task 5.

We balance the training data so that embeddings from all layers are represented equally in each class. To show that the task number is encoded in the model's embeddings and is not an artifact of the classifier's weights, we construct a control setting where, for each class, we generate random vectors with the same dimension, drawn from a normal distribution with the mean and variance of the layer embeddings in that class.[3]

## 3 Results

To show the robustness of our findings, we test pretrained Transformer models on corresponding datasets from three modalities: text, vision and speech.

---

[1] In all of our experiments, we only consider tokens in the input where the 1st saturation layer satisfies that $l^1 \leq 0.85 \cdot N$, to ensure that there are enough layers after it for meaningful analysis.

[2] $l^5$ is ill-defined in this case as the 5-th token doesn't reach saturation before the last layer.

[3] The number of tasks is determined per model to be the maximum number for which after balancing the data there are at least 10 embeddings from each layer in each class from at least 4 different layers.

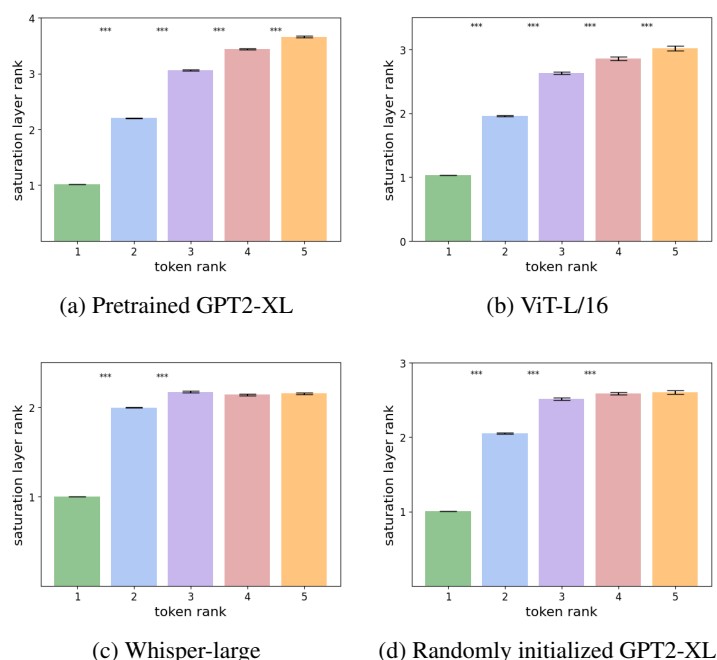

(a) Pretrained GPT2-XL

(b) ViT-L/16

(c) Whisper-large

(d) Randomly initialized GPT2-XL

Figure 3: Average rank of the $k$-th saturation layer among the saturation layers for k=1,..,5 with standard error bars. Asterisks indicate statistically significant differences between consecutive token ranks (*** represents $p < 0.001$), based on an independent samples t-test.

## 3.1 TEXT TRANSFORMER

We first conduct our experiments on a pretrained GPT2-XL model, an auto-regressive decoder-only LLM, using 60K tokens taken from 100 randomly sampled texts from CNN/DM dataset (Hermann et al., 2015). We additionally reproduce our results using Llama3-8B (Dubey et al., 2024) model on MMLU (Hendrycks et al., 2020) and Hellaswag (Zellers et al., 2019) benchmarks (see Appendix B.2).

**Tokens reach saturation in order of ranking.** Figure 8a shows the average rank of the k-th saturation layer for each $k$. This value increases monotonically with $k$, and the difference between each two consecutive token ranks is statistically significant with $p < 0.001$ based on a pairwise independent samples t-test. This supports our claim that saturation events happen sequentially according to token ranking in this LLM. To statistically validate this phenomenon we use a stricter version of Kendall's $\tau$ coefficient, where we also consider ties as disagreements (see Appendix A.1 for details and mathematical formulation). This is done to discount cases where two or more tokens reach saturation at the same layer. The coefficient takes values in the range $[-1, 1]$ where values close to 1 indicate strong agreement, and values close to -1 indicate strong disagreement between the rankings. To check whether the sequence of saturation layers of the top-$k$ tokens $(l^1, .., l^k)$ is strictly increasing, we use that sequence as one ranking, and the sequence $(1, 2, .., k)$ as the other. $k$ is set independently for each token in the input to be the largest token index such that this token's reaches saturation by our definition i.e. $l^k < N$. The average $\tau$ coefficient indicates moderate agreement between the rankings, which is larger than all values over 1K permutations, where the saturation layers sequence were randomly shuffled for each instance, resulting in $p < 0.001$.

**Task number can be predicted from model embeddings.** We split the data into train and test using 5-fold cross validation, and report the mean and standard error of the accuracy. Table 1 shows that the logistic regression classifier trained on embeddings extracted from pretrained GPT2-XL model achieves very high accuracy, while the classifier trained on the random embeddings in the control setting performs approximately at chance level (see Appendix A.3 for accuracy and ROC-AUC scores per class). From this we conclude that the representations of the hidden layers across examples encode task specific information and that the saturation layers as we defined them are the points of transition between those tasks.

Table 1: Accuracy of task number logistic regression classifier showing that in all modalities the layer embeddings contain information about the task number. Asterisks indicate statistically significant accuracy (*** represents $p < 0.001$), based on an Binomial Distribution probability test.

| Model | Layer Embeddings | Random Embeddings | Chance Level |
|---|---|---|---|
| GPT2-XL (pretrained) | **91.4**\*\*\* $\pm$ 0.3 | $20.6 \pm 0.5$ | 20.0 |
| GPT2-XL (random initialization) | **86.1**\*\*\* $\pm$ 0.7 | $32.7 \pm 0.1$ | 33.3 |
| ViT-L/16 (pretrained) | **63.8**\*\*\* $\pm$ 0.1 | $21.0 \pm 0.5$ | 20.0 |
| Whisper-large (pretrained) | **52.7**\*\*\* $\pm$ 0.1 | $24.5 \pm 0.4$ | 25.0 |

## 3.2 VISION TRANSFORMER

Encoder-only image-classification ViTs take as input a sequence of linear projections of equal-sized image patches with added position embedding and a special "class token" denoted [CLS]. Following the work of (Vilas et al., 2024) we use a version of the logit lens adapted to ViT to project the hidden state representations of each layer in the encoder onto the class embedding space using the output embedding matrix. Importantly this is done only for the [CLS] token for each image under the assumption that it best represents the model's prediction, since during ViT's pretraining the only token projected onto the class-embedding space is the [CLS] token from the last layer.

For our experiments we use the ViT-L/16 variant pretrained on ImageNet-21k and fine-tuned on ImageNet 2012, which has 1K classes and 24 layers, and run inference on 5K randomly sampled images from the CIFAR-10 (Krizhevsky et al., 2009) dataset. Figure 3b demonstrates the high correspondence between saturation layer and token rank, and the stricter Kendall's $\tau$ coefficient indicates a moderate agreement between the saturation layers order and the sequence $(1, 2, .., k)$ which is statistically significant with $p < 0.001$ (see Appendix A.1), supporting our claim that in this domain as well as in text the saturation layers are highly ordered. Furthermore, Table 1 shows that the task index can be predicted from the hidden layer activations with high accuracy well above chance and control setting.

## 3.3 SPEECH TRANSFORMER

Whisper is an encoder-decoder Transformer model trained on many different speech processing tasks, including ASR. Although recently there have been attempts to increase efficiency in ASR, such as Malard et al. (2023), the concept of early exit has yet to be explored in this setting, and to the best of our knowledge there has not been work done concerning saturation events in speech models. We adapt the logit lens and apply it *only* to the decoder stack of Whisper-large, which has 32 layers, under the assumption that representations in the encoder stack are inherently different and projecting them onto the token vocabulary space would not be meaningful. For our dataset we randomly sample 5K audios from LibriSpeech (Panayotov et al., 2015).

In addition to reproducing the classical top-1 saturation event established in language and vision models in previous work, we also show in Figure 3c evidence for the tendency of the top-k tokens to reach saturation in order in this model as well, albeit only up to the third token. Moreover, a permutation test performed on the stricter Kendall's $\tau$ coefficient demonstrates that the agreement between the token ranking and the order of saturation layers is statistically significant with $p < 0.001$ (see Appendix A.1). We suspect that the order deteriorates in later tokens due to the fact that each layer in the decoder is conditioned on the last layer of the encoder which may interfere with the task transition mechanism by "blurring the lines" between the tasks. Even so, Table 1 shows that the task index can be predicted from Whisper's decoder layers' embeddings for tasks 1 through 4 with accuracy much higher than chance or that achieved in the control setting.

## 4 ANALYSIS

We have shown that top-k tokens tend to reach saturation in order of their ranking, as well as the plausibility of the underlying task transition mechanism over multiple modalities and variants of Transformers: decoder only, encoder only and full Transformer; in section 4.1 we argue that this

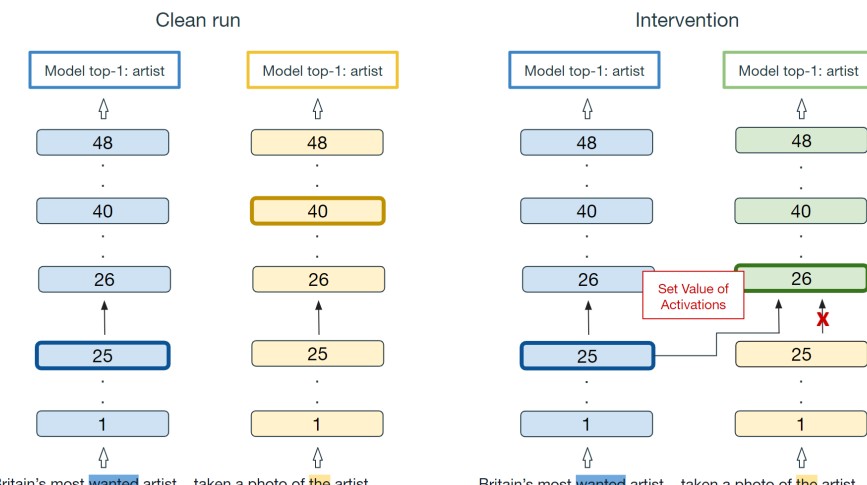

Figure 4: *Left*: Forward pass of two input tokens ("wanted" and "the") in the same context for which the model's final top-1 prediction is the same ("artist"), but the 1st saturation layers are different (25 and 40 respectively). *Right*: By injecting the output from the top-1 saturation layer of "the" as input to the subsequent layer of "artist", we trigger a saturation at the injected layer (26) in the post-intervention run, without altering the top-1 prediction. Saturation layers are marked in bold. The use of activations from adjacent layers is not depicted for the sake of clarity.

phenomenon is inherent to the architecture itself and in section 4.2 we delve deeper into the way the model transitions between tasks, demonstrating that we can cause the model to switch to the next task using an intervention procedure.

### 4.1    UNTRAINED TRANSFORMERS ALSO DETERMINE IN ORDER

We repeat our experiments on an untrained GPT2-XL with randomly initialized weights on the same amount of randomly sampled tokens from CNN/DM dataset as with the pretrained model. Surprisingly, Figure 3d shows that the top-k tokens tend to reach saturation in order up to the 4th token, and although the stricter Kendall's $\tau$ coefficient is lower than in the pretrained GPT2-XL model (see Appendix A.1), it is still statistically significant.

In addition, Table 1 shows that the task transition classifier's accuracy is more than $2.5x$ times higher than chance or that of the control setting. The ability of the classifier to extract the task index from the hidden layers' representations in this setting is especially remarkable, demonstrating that despite the randomness of the weights as well of the identities of the predicted tokens, there is still highly ordered information encoded in the model originating only from the constraints of the architecture.

### 4.2    INTERVENING IN LAYER ACTIVATIONS CAUSES TASK SWITCH

Using the probing analysis, we demonstrated that the tasks, as we defined them, are distinct enough to be separated by a simple classifier, that saturation layers mark the boundaries between them, and that the task index is encoded in the hidden layer embeddings. We argue that in addition, each saturation layer encodes the signal to transition to the next task, and all subsequent layers contain the information that the previous task has been completed and that the relevant token is fixed. This can be thought of as switch being flipped "on" for each token that reaches saturation, and remaining "on" from the saturation layer onwards.

To causally validate this claim, taking inspiration from Stolfo et al. (2023), we perform an intervention (visualized in Figure 4) in which we "inject" the output from the 1st saturation layer of sample $s_1$ as input into the subsequent layer in the run on sample $s_2$ and check how this affects the 1st saturation layer of $s_2$. If these activations contain the signal to switch to the next task, we expect this intervention to cause the model to switch to task 2 at the injected layer in the post-intervention

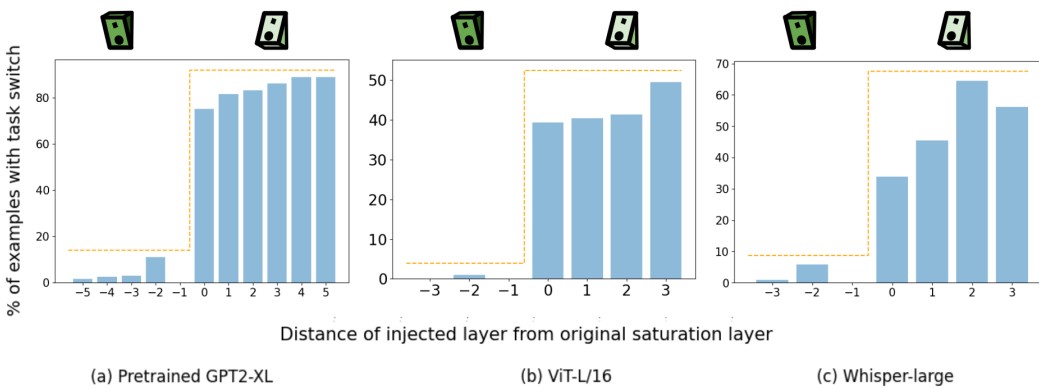

Figure 5: *Flipping the Top-1 Switch.* The percentage of examples where the top-1 saturation occurred at the injected layer after the intervention, shown as a function of the layer from which the injected activations were taken, relative to the original saturation layer (e.g., $-2$ means activations were taken from two layers before the original saturation layer).

run, i.e. in the new post-intervention run the 1st saturation layer should be the one on which the intervention is performed which is $l^1(s_1) + 1$. To minimize the effect of confounding factors, we choose pairs of samples $s_1$ and $s_2$ that share context and where the original top-1 prediction of the model is the same, but there is a big difference in their 1st saturation layers s.t. $l^1(s_1) < l^1(s_2)$. In the example depicted in Figure 4 $s_1 = $ "*wanted*" and $s_2 = $ "*the*", and for both the model's top-1 next-word prediction is "artist", but $l^1(s_1) = 25$ while $l^1(s_2) = 40$. Injecting the output of layer $l^1(s_1)$ into the subsequent layer (26) in the run for $s_2$ should cause the model to switch to task 2, resulting in layer 26 being the new 1st saturation layer post intervention.

Moreover, we would expect the same thing to happen when injecting activations from a layer $l$ *after* the 1st saturation layer, i.e. $l > l^1(s_1)$, since they should contain the information that the top-1 token is fixed. On the other hand, activations from a layer $l'$ *before* the 1st saturation layer, i.e. $l' < l^1(s_1)$ should not result in saturation at the injected layer as the switch is still "off" in our analogy, indicating to the model that it still working on task 1. To test this, we repeat the same steps with activations from 5 layers before and after the 1st saturation layer $[l^1(s_1) - 5, l^1(s_1) + 5]$ each time injecting them as input into the subsequent layer.

Figure 5(a) shows the results of this procedure performed using pretrained GPT2-XL on 200 token pairs taken from 5 randomly sampled texts from the CNN/DM dataset, Figure 5(b) shows similar results reproduced using ViT-L/16 on 200 pairs of images from CIFAR-10 dataset, and Figure 5(c) shows the results of the intervention on Whisper-large on 200 token pairs from 100 randomly sampled audios from LibriSpeech [4]. There is a stark difference in the effect the injected activations have on the 1st saturation layer post-intervention when the activations are taken from the 1st saturation layer in the original run or one of the following layers, compared to the layers before it. When the injected activations are taken from an earlier layer, the new top-1 saturation almost never occurs at the injected layer, whereas when the injected activations are taken from the saturation layer or a later layer the top-1 saturation occurs at the injected layer in a high percentage of cases. This drastic change resembles a step function, and is in line with our description of a switch being flipped "on" at the 1st saturation layer and remaining turned on in all subsequent layers, indicating to the model to switch to the next task and keep the top-1 constant.

## 5 PRACTICAL APPLICATIONS

In this section, we show that our findings can be leveraged for computation efficiency and better performance in LLMs.

---

[4]See Appendix A.4 for a formal description of the procedure as well as details on how we adapted it for vision and speech modalities

Table 2: Highest accuracy and corresponding speedup-ratio achieved by each early-exit strategy.

|  | Softmax Response | State Saturation | Ours |
|---|---|---|---|
| **Accuracy** | $35.9 \pm 0.6$ | $37.5 \pm 0.7$ | $\mathbf{40.0} \pm 0.7$ |
| **Speedup Ratio** | $1.126 \pm 0.004$ | $1.003 \pm 0.001$ | $\mathbf{1.185} \pm 0.009$ |

## 5.1 New Early-Exit Strategy

We propose a new token-level dynamic inference method based on the task-transition classifier described in Section 2.3, where the early exit layer for each token is defined as the earliest layer which is predicted to belong to task 2 by the classifier. The idea being that once the model has transitioned into the second task, it has finished with the first task of determining the top-1 token. To demonstrate the viability and advantages of this method, we compare it to two other local confidence measures introduced by Schuster et al. (2022): softmax response (the difference between the top two values of the logits after softmax) and hidden-state saturation (cosine similarity between two consecutive layer embeddings), both recently found to be competitive with other early exiting methods (Zhou et al., 2024). Since dynamic decoding is not the focus of this paper, we calculate the metrics for each measure while propagating states from the layers after the "early exit" as in regular inference.[5]

Table 17 shows our results on a pre-trained GPT2-XL model and 100 randomly sampled texts from CNN/DM dataset. We evaluate the model on next-word prediction, and compute the speedup ratio for each method as the number of layers it uses for each token divided by the total number of layers in the model, and average across all tokens. For the two other local confidence measures we calculate these metrics at various thresholds (see details in Appendix A.5), while in our measure the class is selected based on the highest predicted score among all classes. Our strategy outperforms the other two when considering the trade-off between next-word prediction accuracy and speedup ratio, and requires no training besides that of a simple logistic regression classifier on a relatively small amount of data. We find that the difference in accuracy between our strategy and the other two methods to be statistically significant with $p < 0.001$ using an independent samples t-test.

## 5.2 Improved Language Modeling

Popular decoding methods in language generation such as top-k (Fan et al., 2018) or top-p (Holtzman et al., 2020) sample the next token according to the shifted distribution induced by probabilities of the top ranking tokens. Based on our task-transition mechanism and the assumption that the tasks represent relevant computation, we argue that top ranking tokens that are determined in the last layer represent less meaningful predictions, since the model only had enough layers for the first task in these instances.

To test this hypothesis we compare the accuracy of the second ranked token in the next word prediction task between two conditions: (1) the second token's saturation layer is at least 7 layers before the output, to increase the chances that this is a "true" saturation as the model had enough layers to change its (recurring error) prediction and it is not due to noise; (2) the second token does not saturate, and is determined only in the last layer (See Appendix B.3 for more details). In both cases we only look at examples where the top-1 token is not the correct prediction. The number of layers in the first setting is a hyper-parameter, and future work should investigate its affects on the second ranked token accuracy.

Using 100 randomly sampled texts from CNN/DM dataset and pretrained GPT2-XL predictions, we find that in the first condition the accuracy is 33.36%, and in the second condition it is only 18.04%. A Two Proportion Z-Test indicates a statistically significant difference between the groups ($p < 0.001$). This supports our claim that top-k tokens that reach true saturation are more plausible than those that are determined only in the last layer, which has potential implications for generation decoding strategies which consider tokens beyond the top-1.

---

[5]This is an informative comparison between the measures, as the effect of a state copying mechanism for skipped layers on model's performance is negligible Schuster et al. (2022).

## 6 RELATED WORK

There are multiple ways of thinking about the role of intermediate layers in Deep Neural Networks (DNNs) in general, and Transformers in particular. The iterative inference hypothesis interprets each layer as an iteration from an iterative and convergent process (Simoulin & Crabbé, 2021), suggesting that each layer incrementally refines the hidden representation by gradually shaping the next token prediction (Geva et al., 2022; Belrose et al., 2023; Rushing & Nanda, 2024). We argue that our findings challenge this view, given the discrete nature of the tasks in the proposed task-transition mechanism and the sharp transitions between them.

Pruning is another approach, focused on mitigating the redundancy inherent to large machine learning models by removing unnecessary parameters. Recent work has applied structured pruning methods to Transformer based LLMs, dropping whole modules, from self-attention layers (Artzy & Schwartz, 2024; He et al., 2024) to full Transformer blocks (Sun et al., 2024; Men et al., 2024). These studies often focus on the middle layers of the model, and claim to reduce memory and computation costs without degrading performance on downstream tasks. It's important to note that these works evaluate the accuracy before and after pruning based only on the top-1 prediction of the model, even though stochastic generation methods such as top-p (Holtzman et al., 2019) and top-k (Fan et al., 2018) are preferable to deterministic decoding in certain settings such as open-ended tasks as they produce more coherent and varied text (Shi et al., 2024). In light of this and of our results regarding the sequential saturation of top ranking tokens, we suggest that future work takes this into account, since what may seem as redundancy is actually necessary computation that is not reflected in the measured metric.

The logit lens has also been used to study intermediate layers in a wide variety of interpretability papers (Yang et al., 2024; Wendler et al., 2024; Halawi et al., 2023). Despite this, Belrose et al. (2023) claim that it can produce implausible results due to the difference in representations between layers. To address this issue they introduce the "tuned lens", in which an affine transformation is learned for each layer in the model with a distillation loss, so that its image under the unembedding matrix matches the final layer logits as closely as possible. Although this method may be better at approximating final top-1 prediction from intermediate layers, our work highlights why this might actually be a disadvantage when attempting to gain insights into the computational process of the Transformer, as it could obscure the changing dynamics of the lower ranked tokens.

## 7 CONCLUSION AND FUTURE WORK

This paper systematically investigates the unexplored question of what computation is performed by the Transformer layers following a top-1 saturation event. We find that the top-k tokens (for $k > 1$) go through similar saturation events in the order of their ranking. We argue that this phenomenon is inherent to the Transformer architecture, replicating our results on an untrained model and demonstrating its robustness over multiple modalities: text, vision, and (to a lesser extent) speech. We then provide evidence in support of a task transition as underlying mechanism for this ordered saturation, showing that we can predict task index from the hidden layers' embeddings, as well as cause the model to switch from the first task to the second via an intervention procedure. Our findings also hold promise in improving inference efficiency and next word prediction accuracy as suggested by the preliminary results in the Practical Applications section.

**Limitations and Future Work** Although our analysis sheds light on the high-level task transition mechanism behind the ordered saturation of top-k tokens, there is still a need for more work to determine which components in the Transformer architecture give rise to it, via ablation studies for example, as well as more concrete explanation for how the model keeps the "chosen" tokens constant after their saturation events across remaining layers. In addition, we did not consider whether the model encountered the data used in our experiments during training as a relevant factor. Finally, as we only explored Transformer architectures, it is necessary to check whether other types of DNNs also determine their top-k tokens in order. Recurrent Neural Networks (RNNs) might be of particular interest due to their mathematical equivalence to decoder only Transformers (Oren et al., 2024), and based on previous work successfully applying the logit lens to them to extract meaningful predictions from intermediate layers (Paulo et al., 2024).

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

# A APPENDIX

## A.1 STRICTER KENDALL'S TAU

We define a version of Kendall's tau coefficient measuring the ordinal association between two tanking, where one-sided ties are considered discordant unlike the regular metric, where ties are typically either ignored or handled as neutral, meaning they neither count as concordant nor discordant.

Given two rankings $x = (x_1, x_2, ., x_n)$ and $y = (y_1, y_2, ., y_n)$, let pair$(i, j)$ be a pair of indices where $1 \leq i < j \leq n$.

We define the pair as *concordant* if the rankings in both sequences agree, meaning:

$$(x_i > x_j \text{ and } y_i > y_j) \quad \text{or} \quad (x_i < x_j \text{ and } y_i < y_j) \quad \text{or} \quad (x_i = x_j \text{ and } y_i = y_j)$$

The pair is *discordant* if:

$$(x_i > x_j \text{ and } y_i < y_j) \quad \text{or} \quad (x_i > x_j \text{ and } y_i > y_j) \quad \text{or} \quad (x_i = x_j \text{ and } y_i \neq y_j) \quad \text{or} \quad (x_i \neq x_j \text{ and } y_i = y_j)$$

The coefficient $\tau_{\text{strict}}$, is computed as:

$$\tau_{\text{strict}} = \frac{C - D}{C + D},$$

where $C$ is the number of concordant pairs, and $D$ is the number of discordant pairs (including ties), ranging in values between $[-1, 1]$.

Table 3 summarizes the results of this metric across the different models discussed in the paper, along with the p-values of the permutation test performed for the mean $\tau_{strict}$ for each.

Table 3: Stricter Kendall's tau coefficients and p-values for each model

| Model | $\tau_{strict}$ (avg $\pm$ ste) | $p_{value}$ | $\tau_{strict} > 0$ |
|---|---|---|---|
| GPT2-XL (pre-trained) | $0.187 \pm 0.004$ | $< 0.001$ | $67.39\%$ |
| GPT2-XL (random initialization) | $0.082 \pm 0.009$ | $< 0.001$ | $49.48\%$ |
| ViT-L/16 (pre-trained) | $0.149 \pm 0.007$ | $< 0.001$ | $58.94\%$ |
| Whisper-large (pre-trained) | $0.210 \pm 0.009$ | $< 0.001$ | $63.78\%$ |

## A.2 TRAINING DATA FOR TASK TRANSITION CLASSIFIER

Table 4 shows from which layers we took embeddings to train the task-transition classifier for each model.

Table 4: Task transition probing data

| Model | Layers | Dataset size |
|---|---|---|
| GPT2-XL (pre-trained) | $23 - 40$ | $6K$ |
| GPT2-XL (random initialization) | $31 - 41$ | $2K$ |
| ViT-L/16 (pre-trained) | $16 - 21$ | $2K$ |
| Whisper-large (pre-trained) | $29 - 32$ | $4K$ |

## A.3 PER CLASS METRICS FOR TASK TRANSITION CLASSIFIER

Table 5 shows accuracy scores per-class for each model, while Table 6 shows ROC-AUC scores per-class for each model.

Table 5: Task transition probing per-class accuracy scores

| Model | Task 1 | Task 2 | Task 3 | Task 4 | Task 5 |
|---|---|---|---|---|---|
| GPT2-XL (pre-trained) | 0.862 | 0.92 | 0.926 | 0.921 | 0.941 |
| GPT2-XL (random initialization) | 0.782 | 0.83 | 0.972 | — | — |
| ViT-L/16 (pre-trained) | 0.646 | 0.618 | 0.603 | 0.644 | 0.678 |
| Whisper-large (pre-trained) | 0.520 | 0.528 | 0.484 | 0.576 | — |

Table 6: Task transition probing per-class ROC-AUC scores

| Model | Task 1 | Task 2 | Task 3 | Task 4 | Task 5 |
|---|---|---|---|---|---|
| GPT2-XL (pre-trained) | 0.966 | 0.987 | 0.974 | 0.974 | 0.985 |
| GPT2-XL (random initialization) | 0.929 | 0.951 | 0.995 | — | — |
| ViT-L/16 (pre-trained) | 0.855 | 0.821 | 0.809 | 0.829 | 0.866 |
| Whisper-large (pre-trained) | 0.777 | 0.767 | 0.695 | 0.765 | — |

### A.4 INTERVENTION PROCEDURE ADDITIONAL DETAILS

Formally, this procedure consists of the following steps:

1. Given an input sequence $x = <x_1, ..., x_t>$ we first pass it through the model as in regular inference while storing the activation values at all hidden layers, i.e $h_i^l$ for all $1 \leq i \leq t$, $1 \leq l \leq N$.

2. We calculate the saturation layer $l_i^1$ of the 1st token for each token $w_i$ in the text.

3. We sample pairs of token indexes $i, j$ in the text that the satisfy the following conditions:

   (a) The distance between $i$ and $j$ is no more than 40 tokens, i.e. $|i - j| \leq 40$.
   This is a precaution to minimize the effect of the difference in context on the model's predictions after intervention.

   (b) The model's top-1 prediction (in the final layer) for both indexes is the same token y, meaning $y = argmax(softmax(Eh_i^N)) = argmax(softmax(Eh_j^N))$.
   The goal here is to avoid a conflict in the top-1 predictions which could be a confounding factor.

   (c) There is a difference of at least 10 layers between the 1st token saturation layers of i and j, such that $|l_i^1 - l_j^1| \leq 10$, to ensure that the change in saturation layer after intervention is significant.

   For convenience's sake we will assume in the remainder of the procedure description that $l_i^1 < l_j^1$, i.e that the saturation layer of the first index in the pair is smaller then that of the second index (even though both cases are allowed by our conditions).

4. We perform 11 additional forward passes, each time injecting the output from layer $l'$ in range $[l_i^1 - 5, l_i^1 + 5]$ at position $i$ as input into layer $l' + 1$ at position j . The goal here is to to quantify the difference in effect between layers preceding the saturation event and those after it.

5. We measure the causal effect of the intervention by calculating the percent of examples where the saturation layer of the 1st token after intervention $\tilde{l}_j^1$ is the layer on which we intervened, i.e. $\tilde{l}_j^1 = l + 1$.

For example, in the setting depicted in 4 we would take the indexes of the marked tokens "wanted" and "the" as our pair, where the original top-1 prediction in both is "artist". The top-1 saturation layer in the clean run for "wanted" is layer 25, so we would inject activations from layers 20 to 30 one at a time as inputs into the corresponding subsequent layers in the run of token "the" (i.e. layers

21 to 31), and check for each one if the injected layer became the new top-1 saturation layer after the intervention.

### A.4.1 Intervention Procedure on ViT

To adapt the intervention procedure described in section 4.2 to ViT-L/16 and the image classification setting we made the following modifications:

1. Since each image is processed independently by the model there is no need for two images to share a context, so the only requirements for two images to be chosen as a relevant pair were: a distance of at least 5 layers (instead of 10, since this model only has 24 total layers compared to GPT2-XL's 48) between the top-1 saturation layers, and the same top-1 class prediction in the final layer.

2. For each image, as in all experiments conducted on this model we only consider the prediction at index 0 corresponding to the [CLS] token in the input.

3. We used embeddings from 3 layers before and 3 layers after the saturation layer (instead of 5, again due to the smaller number of layers) resulting in 7 total forward passes.

Figure 5 shows that the results for this model follow a similar step function pattern to the ones for GPT2, where injecting embeddings from the top-1 saturation layer or one of the subsequent layers causes the model to "immediately" (at the injected layer) switch to the second task in a high percentage of cases, when compared to injecting embeddings from one of the layer before the top-1 saturation which almost never has the same effect.

### A.4.2 Intervention Procedure on Whisper

We made the following adjustments to run the procedure described in section 4.2 on Whisper-large and 200 token pairs from randomly sampled 50 audios from the LibriSpeech dataset:

1. Since the average audio in LibriSpeech is 10 seconds long there are not enough tokens in one sample to find relevant pairs, so we wave the requirement for a pair to share context and only leave two conditions: a distance of at least 5 layers (instead of 10, since this model only has 32 total layers compared to GPT2-XL's 48) between the top-1 saturation layers, and the same top-1 prediction in the final layer.

2. We used embeddings from 3 layers before and 3 layers after the saturation layer (instead of 5, again due to the smaller number of layers) resulting in 7 total forward passes.

Figure 5 shows that the results for this model follow a similar pattern to the other two models, even though the effect increases in the following layers after the saturation event.

### A.5 Additional Details for Token-Level Early Exit Measures Comparison

Table 7 shows the accuracy and speedup ratio of the Softmax Response token-level early-exit strategy at various thresholds.

Table 8 shows the accuracy and speedup ratio of the Hidden-state saturation token-level early-exit strategy at various thresholds.

Figure 14 visualizes the performance-efficiency trade-off of both methods in comparison to our novel early-exit strategy, as well as baseline and oracle.

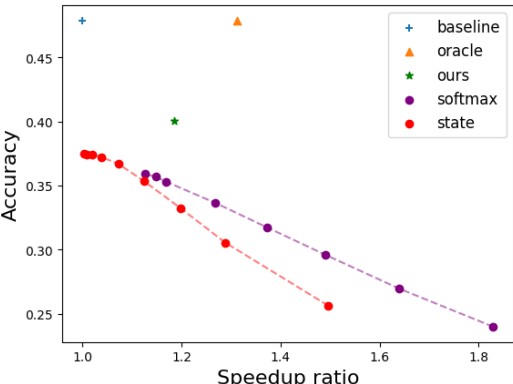

Figure 6: Performance-efficiency trade-off comparison of different confidence measures against a static baseline (where all layers are used for each token) and a local oracle measure (where the early exit is at the top-1 saturation layer). The graph shows softmax and state confidence measure results at different thresholds. Our method achieves the highest next-word prediction accuracy out of all early-exist methods while providing significant speedup compared to the baseline.

Table 7: Softmax Response accuracy & speedup ratios at various confidence thresholds

| | Thresholds | | | | | | | |
| --- | --- | --- | --- | --- | --- | --- | --- | --- |
| | 0.4 | 0.5 | 0.6 | 0.7 | 0.8 | 0.9 | 0.92 | 0.94 |
| **Accuracy** | 0.240 | 0.270 | 0.296 | 0.317 | 0.336 | 0.353 | 0.357 | 0.359 |
| **Speedup Ratio** | 1.830 | 1.640 | 1.491 | 1.373 | 1.269 | 1.169 | 1.148 | 1.126 |

Table 8: Hidden-sate saturation accuracy & speedup ratios at various confidence thresholds

| | Thresholds | | | | | | | |
| --- | --- | --- | --- | --- | --- | --- | --- | --- |
| | 0.986 | 0.988 | 0.989 | 0.99 | 0.991 | 0.992 | 0.993 | 0.994 | 0.995 |
| **Accuracy** | 0.256 | 0.306 | 0.333 | 0.353 | 0.367 | 0.372 | 0.374 | 0.374 | 0.375 |
| **Speedup Ratio** | 1.496 | 1.288 | 1.197 | 1.125 | 1.073 | 1.039 | 1.019 | 1.008 | 1.003 |

## B SUPPLEMENTARIES BASED ON REVIEWER FEEDBACK

### B.1 SATURATION STATISTICS

Using GPT2-XL model and 200 randomly chosen texts from CNN/DM dataset we show that saturation events are common for top-k tokens in Table 9, with over 80% of samples reaching top-1 saturation at least 3 layers before output. Even if we consider only cases where top-1 saturation happens in the first 85% layers of the model, as we do in all of our experiments in Section 2.2, we find that this includes 51.5% of all samples.

Table 9: Percent of samples where top-k tokens reach saturation at least 3 layers before output

| Model | 1st token | 2nd token | 3d token | 4th token | 5th token |
|---|---|---|---|---|---|
| GPT2-XL (pre-trained) | 80.7% | 56.6% | 40.2% | 30.1% | 23.6% |

We show in Figure 7 that the samples that reach top-1 saturation belong to all different parts of speech (POS), and are not just function words for example, with over 27% of them being nouns.

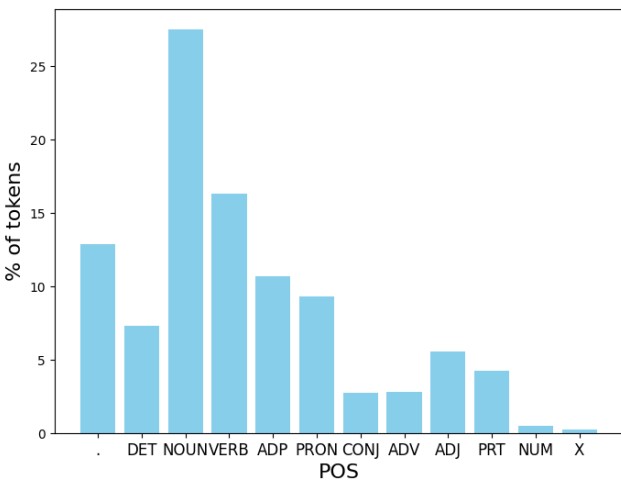

Figure 7: POS of samples that reach top-1 saturation in first 85% of layers of GPT2-XL

### B.2 LLAMA3 RESULTS

We reproduce the qualitative results from Sections 3.1 and 4.2 as well as the practical applications from Section 5 using a 8-bit quantized version of pre-trained Llama3-8B (Wendler et al., 2024) model, a multilingual SOTA decoder-only LLM, on two datasets: MMLU (Hendrycks et al., 2020) and HellaSwag (Zellers et al., 2019).

MMLU is a multitask benchmark consisting of multiple-choice questions from 57 different subjects including elementary mathematics, US history, computer science, law, and more. We use 1K randomly sampled questions from the MMLU test split. We employ the following prompt to present each task's questions, answer choices, and correct answer, ensuring a uniform input structure.

**Prompt format:**

Question: <QUESTION>
  A. <CHOICE A>
  B. <CHOICE B>
  C. <CHOICE C>

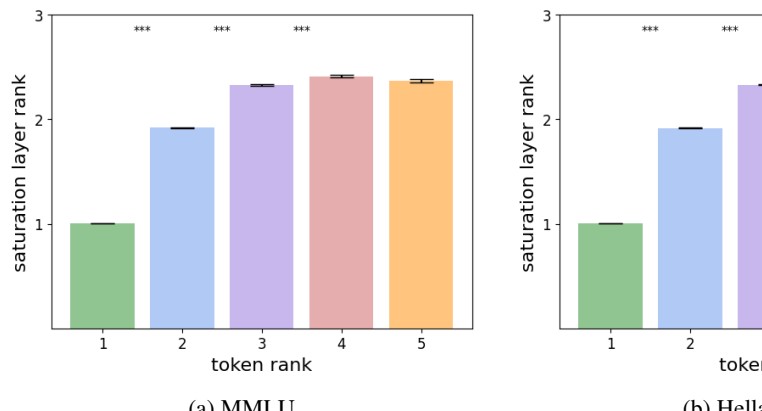

(a) MMLU                    (b) Hellaswag

Figure 8: Average rank of the $k$-th saturation layer among the saturation layers for k=1,..,5 of Llama3-8B with standard error bars. Asterisks indicate statistically significant differences between consecutive token ranks (*** represents $p < 0.001$), based on an independent samples t-test.

       D. `<CHOICE D>`
    Answer: `<ANSWER>`

HellaSwag is a commonsense reasoning benchmark, consisting of multiple choice questions where the options are different possible continuations for a given context and the challenge is to choose the most likely one. We use 1K randomly sampled questions from the HellaSwag validation split. We employ the following prompt to systematically present each task's context, continuation options, and correct option, ensuring a uniform input structure.

**Prompt format:**

    Context: `<CONTEXT>`
    Options:
      A. `<OPTION A>`
      B. `<OPTION B>`
      C. `<OPTION C>`
      D. `<OPTION D>`
    The most likely option is: `<ANSWER>`

We show in Figure 8 that the Llama3's top-k tokens reach saturation in order of their ranking up to (and including) the 4th ranking token. We suspect that the phenomenon doesn't extend to the 5th ranking token because this model has only 32 layers compare to the 48 of GPT2-XL.

We additionally validate this agreement between token ranking and saturation layer ranking using a stricter version of Kendall's $\tau$ metric as described in Section 3.1, and find that the average $\tau$ value over 500 questions randomly sampled from MMLU dataset is $0.026 \pm 0.007$, which is statistically significant with $p < 0.001$ based on a random permutation test.

Furthermore, in support of our task transition mechanism, using embeddings extracted from inference over 500 questions randomly sampled from MMLU dataset, we demonstrate that task number can be predicted from Llama3 embeddings. The logistic regression classifier trained over 5K embedding, balanced between classes as described in Section 2.3 achieves average accuracy of 88.1 over 3 classes. We report full results and control settings in Table 10. Asterisks indicate statistically significant accuracy (*** represents $p < 0.001$), based on an Binomial Distribution probability test.

Finally, we show in Figure 9 the results of the intervention procedure described in Section 4.2 using Llama3-8B over 200 token pairs extracted from 10 randomly sampled texts from CNN/DM dataset[6]. Similarly to what we find in Section 4.2, when the injected activations are taken from the top-1

---

[6]We use texts from CNN and not MMLU for this experiment as they tend to be longer and have more pairs that fit our criteria for intervention

Table 10: LLaMA3-8B task transition probing

| Model | Layers | Dataset size | Accuracy | | |
|-------|--------|--------------|----------|---|---|
| | | | Layer embeddings | Random embeddings | Chance |
| LLaMA3-8B (pre-trained) | $19 - 27$ | $5K$ | $\mathbf{88.1}^{***} \pm 0.4$ | $33.1 \pm 0.5$ | $33.3$ |

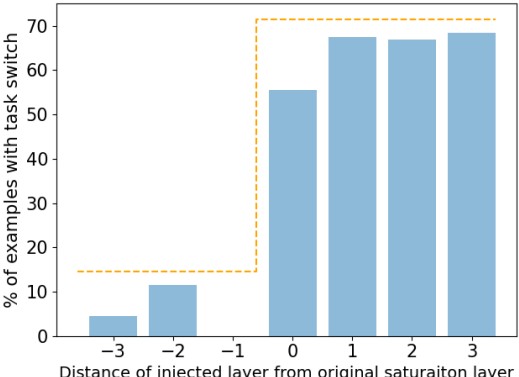

Figure 9: Flipping the Top-1 Switch in Llama3-8B. The percentage of examples where the top-1 saturation occurred at the injected layer after the intervention, shown as a function of the layer from which the injected activations were taken, relative to the original saturation layer (e.g., $-2$ means activations were taken from two layers before the original saturation layer).

Table 11: Llama3-8B: Highest accuracy and corresponding speedup-ratio achieved by each early-exit strategy.

| | **Softmax Response** | **State Saturation** | **Ours** |
|---|---|---|---|
| **Accuracy** | $35.9 \pm 0.6$ | $37.5 \pm 0.7$ | $\mathbf{40.0} \pm 0.7$ |
| **Speedup Ratio** | $1.126 \pm 0.004$ | $1.003 \pm 0.001$ | $\mathbf{1.185} \pm 0.009$ |

saturation layer or later layers, the new top-1 saturation happens at the injected layer much more frequently than when injecting activations from earlier layers, indicating that these layer contain the signal to switch to the next task and keep the top-1 constant.

### B.2.1 LLAMA3 PRACTICAL APPLICATIONS

Table 11 shows our results on a pre-trained Mistral-7B model and 100 randomly sampled texts from CNN/DM dataset. We evaluate the different early-exit strategies as described in Section 5.1. Our strategy outperforms the other two when considering the trade-off between next-word prediction accuracy and speedup ratio, and requires no training besides that of a simple logistic regression classifier on a relatively small amount of data. We find that the difference in accuracy between our strategy and the other two methods to be statistically significant with $p < 0.001$ using an independent samples t-test.

Figure 10 visualizes the performance-efficiency trade-off of both methods in comparison to our novel early-exit strategy, as well as baseline and oracle.

We repeat the same experiment described in Section 5.2, using 100 randomly sampled texts from CNN/DM dataset and pretrained Mistral-7B predictions, varying how many layers before output saturation occurs for the 2nd ranking token. Using a Two Proportion Z-Test we find that difference in next-word prediction accuracy between 2nd token achieving saturation $i$ layers before final layer (with $2 \leq i \leq 6$) condition and the condition of the 2nd token being determined only in the last

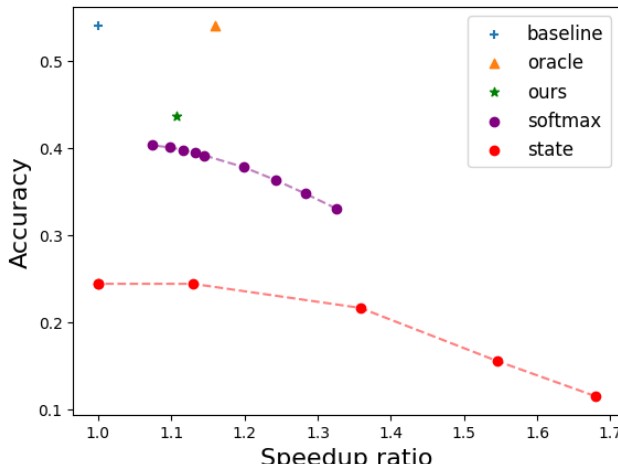

Figure 10: Llama3-8B: Performance-efficiency trade-off comparison of different confidence measures against a static baseline (where all layers are used for each token) and a local oracle measure (where the early exit is at the top-1 saturation layer). The graph shows softmax and state confidence measure results at different thresholds. Our method achieves the highest next-word prediction accuracy out of all early-exist methods while providing significant speedup compared to the baseline.

layer is statistically significant with $p < 0.001$ even when correcting for multiple comparisons. See Table 12 for full results.

Table 12: Llama3-8B: Accuracy of next word prediction of 2nd ranking token when top-1 token is incorrect (comparing no saturation to saturation $i$ layers before output)

| No saturation | Saturation $i$ layers before output | | | | |
|---|---|---|---|---|---|
| | $i = 2$ | $i = 3$ | $i = 4$ | $i = 5$ | $i = 6$ |
| 25.22 | 38.10 | 48.58 | 37.55 | 37.58 | 33.01 |

### B.3 HYPER-PARAMETER VARIATIONS ON IMPROVED LANGUAGE MODELING

We repeat the same experiment described in Section 5.2, varying how many layers before output saturation occurs for the 2nd ranking token. Using a Two Proportion Z-Test we find that difference in next-word prediction accuracy between 2nd token achieving saturation $i$ layers before final layer (with $2 \leq i \leq 10$) condition and the condition of the 2nd token being determined only in the last layer is statistically significant with $p < 0.001$ even when correcting for multiple comparisons. See Table 13 for full results.

Table 13: Accuracy of next word prediction of 2nd ranking token when top-1 token is incorrect (comparing no saturation to saturation $i$ layers before output)

| No saturation | Saturation $i$ layers before output | | | | | | | | |
|---|---|---|---|---|---|---|---|---|---|
| | $i = 2$ | $i = 3$ | $i = 4$ | $i = 5$ | $i = 6$ | $i = 7$ | $i = 8$ | $i = 9$ | $i = 10$ |
| 18.04 | 26.11 | 28.2 | 29.48 | 31.35 | 32.16 | 33.36 | 34.07 | 35.28 | 35.7 |

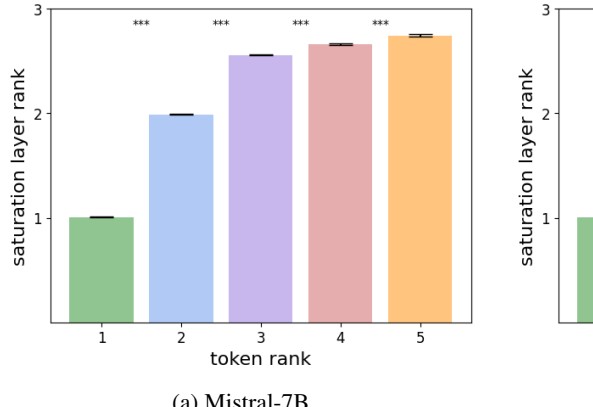 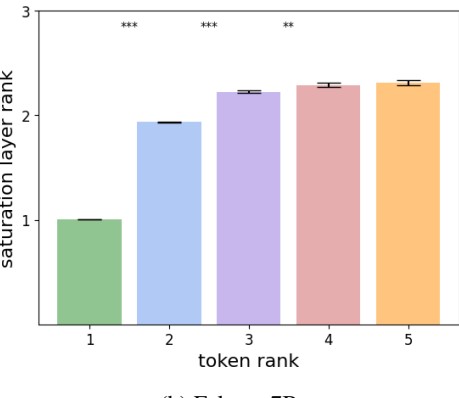

(a) Mistral-7B         (b) Falcon–7B

Figure 11: Average rank of the $k$-th saturation layer among the saturation layers for k=1,..,5 of Mistral-7b and Falcon-7B with standard error bars. Asterisks indicate statistically significant differences between consecutive token ranks (*** represents $p < 0.001$, ** represents $p < 0.01$), based on an independent samples t-test.

### B.4 MISTRAL & FALCON RESULTS

We reproduce our results from Sections 3.1 and 4.2 with 8-bit quantized versions of Mistral-7B (Jiang et al., 2023) and Falcon-7B (Almazrouei et al., 2023) models, both decoder-only LLMs.

Using randomly sampled 1K questions from MMLU dataset for each model and the prompt format described in Section B.2, we show in Figure 11 that the Mistral's top-k tokens reach saturation in order of their ranking up to (and including) the 5th ranking token, while Falcon's top-k tokens reach saturation in order of their ranking up to the 4th token.

We additionally validate this agreement between token ranking and saturation layer ranking using a stricter version of Kendall's $\tau$ metric as described in Section 3.1, and find that the average $\tau$ value over 500 questions randomly sampled from MMLU dataset is $0.08 \pm 0.003$ for Mistral and $0.04 \pm 0.001$, both of which are statistically significant with $p < 0.001$ based on a random permutation test.

Furthermore, in support of our task transition mechanism, using embeddings extracted from inference over 500 questions randomly sampled from MMLU dataset, we demonstrate that task number can be predicted from both Mistral and Falcon hidden layers' embeddings. For Mistral, a logistic regression classifier trained over 3K embeddings, balanced between classes as described in Section 2.3 achieves average accuracy of 85.8 over 4 classes. For Mistral, a logistic regression classifier trained over 2K embeddings, balanced between classes as described in Section 2.3 achieves average accuracy of 91.0 over 4 classes. We report full results and control settings in Table 14. Asterisks indicate statistically significant accuracy (*** represents $p < 0.001$), based on an Binomial Distribution probability test.

Table 14: Mistral and Falcon task transition probing

| Model | Layers | Dataset size | Accuracy | | |
|---|---|---|---|---|---|
| | | | Layer embeddings | Random embeddings | Chance |
| Mistral-7B (pre-trained) | $22 - 27$ | $2K$ | **85.8**\*\*\* $\pm 0.01$ | $27.1 \pm 0.01$ | 25 |
| Falcon-7B (pre-trained) | $19 - 27$ | $2K$ | **91.0**\*\*\* $\pm 0.01$ | $24.6 \pm 0.01$ | 25 |

Finally, we show in Figure 12 the results of the intervention procedure described in Section 4.2 using Mistral-7B and Falcon-7B models over 200 token pairs (each) extracted from 10 randomly sampled texts from CNN/DM dataset. As with the other models, when the injected activations are taken from the top-1 saturation layer or later layers, the new top-1 saturation happens at the injected

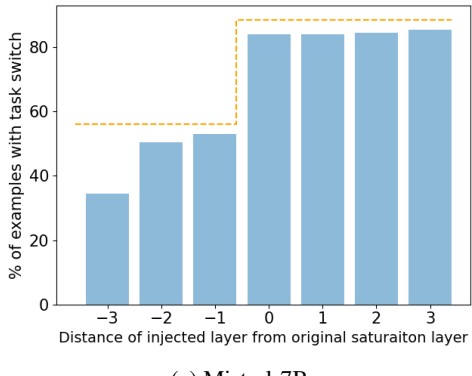
(a) Mistral-7B

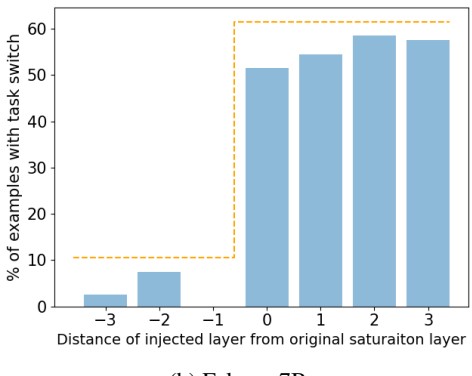
(b) Falcon–7B

Figure 12: Flipping the Top-1 Switch. The percentage of examples where the top-1 saturation occurred at the injected layer after the intervention, shown as a function of the layer from which the injected activations were taken, relative to the original saturation layer (e.g., $-2$ means activations were taken from two layers before the original saturation layer).

Table 15: Mistral-7B: Highest accuracy and corresponding speedup-ratio achieved by each early-exit strategy.

|  | **Softmax Response** | **State Saturation** | **Ours** |
| --- | --- | --- | --- |
| **Accuracy** | $35.9 \pm 0.6$ | $37.5 \pm 0.7$ | **40.0** $\pm 0.7$ |
| **Speedup Ratio** | $1.126 \pm 0.004$ | $1.003 \pm 0.001$ | **1.185** $\pm 0.009$ |

layer much more frequently than when injecting activations from earlier layers, indicating that these layer contain the signal to switch to the next task and keep the top-1 constant.

### B.4.1 MISTRAL PRACTICAL APPLICATIONS

Table 15 shows our results on a pre-trained Mistral-7B model and 100 randomly sampled texts from CNN/DM dataset. We evaluate the different early-exit strategies as described in Section 5.1. Our strategy outperforms the other two when considering the trade-off between next-word prediction accuracy and speedup ratio, and requires no training besides that of a simple logistic regression classifier on a relatively small amount of data. We find that the difference in accuracy between our strategy and the other two methods to be statistically significant with $p < 0.001$ using an independent samples t-test.

Figure 14 visualizes the performance-efficiency trade-off of both methods in comparison to our novel early-exit strategy, as well as baseline and oracle.

We repeat the same experiment described in Section 5.2, using 100 randomly sampled texts from CNN/DM dataset and pretrained Mistral-7B predictions, varying how many layers before output saturation occurs for the 2nd ranking token. Using a Two Proportion Z-Test we find that difference in next-word prediction accuracy between 2nd token achieving saturation $i$ layers before final layer (with $2 \le i \le 6$) condition and the condition of the 2nd token being determined only in the last layer is statistically significant with $p < 0.001$ even when correcting for multiple comparisons. See Table 16 for full results.

### B.4.2 FALCON PRACTICAL APPLICATIONS

Table 17 shows our results on a pre-trained Falcon-7B model and 100 randomly sampled texts from CNN/DM dataset. We evaluate the different early-exit strategies as described in Section 5.1. Our strategy outperforms softmax response when considering the trade-off between next-word prediction accuracy and speedup ratio, and requires no training besides that of a simple logistic regression classifier on a relatively small amount of data. We find that the difference in accuracy between our

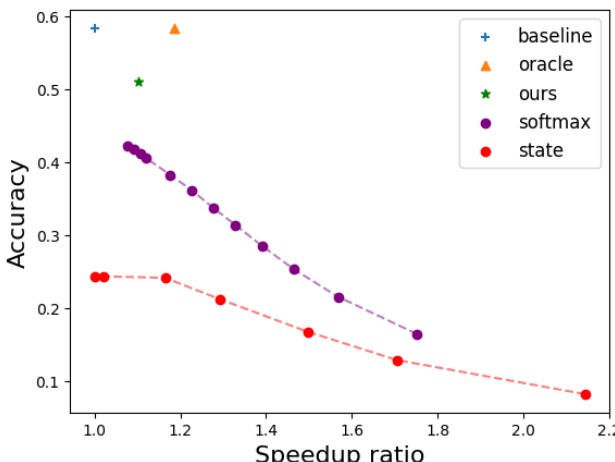

Figure 13: Mistral-7B: Performance-efficiency trade-off comparison of different confidence measures against a static baseline (where all layers are used for each token) and a local oracle measure (where the early exit is at the top-1 saturation layer). The graph shows softmax and state confidence measure results at different thresholds. Our method achieves the highest next-word prediction accuracy out of all early-exist methods while providing significant speedup compared to the baseline.

Table 16: Mistral-7B: Accuracy of next word prediction of 2nd ranking token when top-1 token is incorrect (comparing no saturation to saturation $i$ layers before output)

| No saturation | Saturation $i$ layers before output | | | | |
|---|---|---|---|---|---|
| | $i = 2$ | $i = 3$ | $i = 4$ | $i = 5$ | $i = 6$ |
| 13.812 | 33.37 | 38.43 | 41.03 | 41.91 | 39.15 |

Table 17: Falcon-7B: Highest accuracy and corresponding speedup-ratio achieved by each early-exit strategy.

| | Softmax Response | State Saturation | Ours |
|---|---|---|---|
| **Accuracy** | $35.9 \pm 0.6$ | $37.5 \pm 0.7$ | $\mathbf{40.0} \pm 0.7$ |
| **Speedup Ratio** | $1.126 \pm 0.004$ | $1.003 \pm 0.001$ | $\mathbf{1.185} \pm 0.009$ |

strategy and softmax response to be statistically significant with $p < 0.001$ using an independent samples t-test.

Figure 14 visualizes the performance-efficiency trade-off of both methods in comparison to our novel early-exit strategy, as well as baseline and oracle.

We repeat the same experiment described in Section 5.2, using 100 randomly sampled texts from CNN/DM dataset and pretrained Falcon-7B predictions, varying how many layers before output saturation occurs for the 2nd ranking token. Using a Two Proportion Z-Test we find that difference in next-word prediction accuracy between 2nd token achieving saturation $i$ layers before final layer (with $2 \leq i \leq 6$) condition and the condition of the 2nd token being determined only in the last layer is statistically significant with $p < 0.001$ even when correcting for multiple comparisons. See Table 18 for full results.

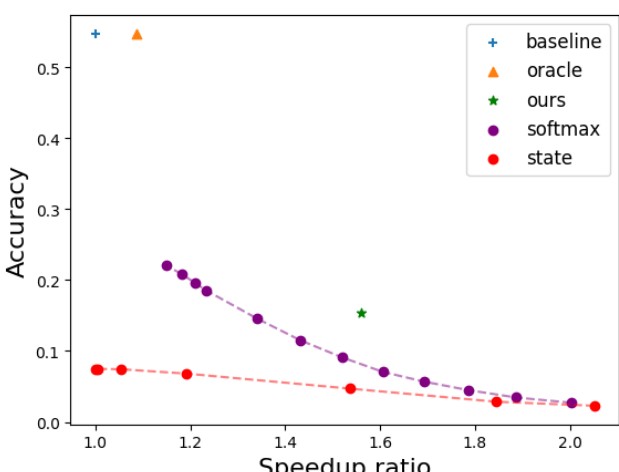

Figure 14: Falcon-7B: Performance-efficiency trade-off comparison of different confidence measures against a static baseline (where all layers are used for each token) and a local oracle measure (where the early exit is at the top-1 saturation layer). The graph shows softmax and state confidence measure results at different thresholds. Our method beats the hidden state strategy, but performs worse than softmax.

Table 18: Falcon-7B: Accuracy of next word prediction of 2nd ranking token when top-1 token is incorrect (comparing no saturation to saturation $i$ layers before output)

| No saturation | Saturation $i$ layers before output | | | | |
|---|---|---|---|---|---|
| | $i = 2$ | $i = 3$ | $i = 4$ | $i = 5$ | $i = 6$ |
| 12.78 | 37.81 | 34.55 | 34.83 | 35.33 | 37.74 |

