# OpenReview forum: "Looking Beyond the Top-1: Transformers Determine Top Tokens in Order"
_ICLR.cc/2025/Conference — Submitted to ICLR 2025_

### Official Review · Reviewer_u1oC · 2024-10-30

**Soundness:** 1
**Presentation:** 2
**Contribution:** 2
**Rating:** 3
**Confidence:** 4

**Summary:**

The paper explores hidden states of intermediate layers of transformer-based models by mapping them onto the vocabulary space to extract ranking over tokens and analyzing their changes over the layers. The authors claim to identify sequential saturation events for top-k tokens, a phenomenon observed across language, vision, and speech models. They propose a type of discrete transition occurs during decoding.

**Strengths:**

Strengths:
The paper emphasizes the importance of investigating layer-wise states transition. The study tackles an important and interesting research question.

**Weaknesses:**

The hypothesis of “saturation events” is taken as granted without sufficient evidence. The mechanism in producing top-1 tokens in LLMs is influenced by various factors, including LLM architecture, contexts, and how an LLM reacts to an input. Based on practice shared by many colleagues in probing LLMs, transformers often alternate their top-1 token predictions, even in later layers, which undermines the claimed findings by this study.

The scope of the experiment is limited in terms of both the number and types of LLMs and tasks. This raises concerns about the generality of the findings. It is important to investigate both NLU and NLG tasks on a diverse range of LLMs under different regimes (e.g., pre-trained, post-trained, decoder-only, encoder-decoder) to ensure comprehensive insights. Instead of limiting to a small-scale LLM like GPT2-XL, the authors may consider exploring state-of-the-art (SOTA) LLMs, such as the LLAMA series, to broaden the scope of their research and enhance the robustness of the conclusions.

The use of ranks and central tendency in Figure 3 is confusing (i.e., why averaging ranks and std in figure 3). There are existing statistic rank tests the authors may leverage. The claimed sequential saturation pattern should be convincingly demonstrated.

The practical application of intervention is unclear, and the results shown in table 2 are not statistically significant. When you choose "Highest accuracy", how do you calculate it? Is it based on EM?

**Questions:**

as above

---

> ### Author Response · Authors · 2024-11-15
>
> We’re glad that you find our research question important and interesting.
>
> Although, to the best of our knowledge, the existence of “saturation events” is not controversial in the literature [1,2,3], we agree that we should have provided some evidence demonstrating their frequency in the data. For this reason, we’ve added a section (currently in Appendix B.1) showing that in GPT2, saturation events are common for top-k tokens, with over 80% of samples reaching top-1 saturation. Even if we consider only cases where top-1 saturation happens in the first 85% layers of the model, as we do in all of our experiments, we find that this includes 51.5% of all samples.
>
> To facilitate your evaluation, we have also uploaded a Jupyter notebook to the supplementary materials called “saturation_events.ipynb”. This notebook contains code which allows you to verify the existence of saturation events as described in the paper on Llama3 model.
>
> Regarding the robustness of our findings, we have reproduced our results in different regimes and architecture types and even across modalities - pre-trained and untrained (randomly initialized) GPT2 which is decoder-only, encode-only pre-trained ViT, and encoder-decoder pre-trained Whisper (see Sections 3.2 and 3.3).
>
> To address your concern of our results generalizing to SOTA models and different NLG tasks, we have reproduced our order saturation and task transition probing experiments using Llama3-8B model on MMLU and Hellaswag benchmarks. We show that Llama3’s top-k tokens reach saturation in order of their ranking up to (and including) the 4th ranking token, and that we’re able to predict the task number from layer embeddings with 88% accuracy. Please see Section B.2 in the appendix for more details.
>
> We show that the phenomenon of ordered saturation is statistically significant using a stricter version of a well known ranking metric Kendall’s tau. If this is not sufficient, could you please suggest which other statistical rank tests we should run?
>
> The goal of the intervention experiment is not to demonstrate a practical application for our findings, but rather to show causality and further validate our proposed task transition mechanism.
>
> Regarding Table 2, thank you for bringing this to our attention. After running an independent samples t-test we find that the difference in accuracy between our method and each of the alternatives is statistically significant, with p < 0.001 and make note of it in Section 5.1.
>
> In Section 5.1 we report the highest accuracy achieved across all tested thresholds for Softmax Response and State Saturation methods, which is calculated for the next-word prediction task as the percent of samples where the predicted top-1 token matches exactly the next token in the text. Please see Section A.7 in the Appendix for further details.
>
> If there are specific models, benchmarks, or analyses you would like us to explore to strengthen the work further, please do not hesitate to let us know.
>
> 1 - Transformer feed-forward layers build predictions by promoting concepts in the vocabulary space. (https://arxiv.org/abs/2203.14680)
>
> 2 - Confident adaptive language modeling. (https://proceedings.neurips.cc/paper_files/paper/2022/hash/6fac9e316a4ae75ea244ddcef1982c71-Abstract-Conference.html)
>
> 3 - Language Models Implement Simple Word2Vec-style Vector Arithmetic (https://arxiv.org/abs/2305.16130)

---

> > ### Author Response · Authors · 2024-11-20
> >
> > Dear Reviewer,
> >
> > Thank you for your feedback on our submission. We have carefully addressed all your comments and provided detailed responses to each of your concerns. In addition, we’ve reproduced our ordered saturation, task transition probing and task switching intervention results on  ***Mistral*** and  ***Falcon*** models (see Section B.4 in the Appendix) to further show the generality of our findings. We hope that our replies clarify the points raised and strengthen the case for our paper.
> >
> > As the discussion period concludes on ***November 26***, we would greatly appreciate any further feedback, comments, or follow-up questions you may have. Your input is invaluable in helping us improve our work.
> >
> > Thank you once again for your time and effort.

---

> > > ### Comment · Reviewer_u1oC · 2024-11-28
> > >
> > > I appreciate the authors' detailed response. However, after careful consideration, I remain unconvinced that the controlled experiments adequately capture the decoding complexity of LLMs in real-world scenarios, where confounding factors such as prompts, contexts, and inner polysemanticity play significant roles. I am concerned the optimistic view of “ordered saturation” may be misleading. It is hard for me to raise the score at this stage. To unveil the mechanism of LLMs' decoding behaviors, I believe a more comprehensive analysis of inner feature interactions, circuits, and external influences is needed, instead of relying solely on oversimplified experiments presented in this submission.

---

> > > > ### Author Response · Authors · 2024-11-28
> > > >
> > > > We appreciate the reviewer’s time and effort in contributing to the discussion.
> > > >
> > > > However we are perplexed by the continuous disregard of the empirical results establishing the phenomenon of ordered saturation events, including the generality of our findings as we show it using multiple LLMs (GPT-2, Llama3, Mistral, Falcon), over multiple domains (text, vision and audio) and even in a randomly initialized Transformer! In addition we uploaded code to the supplementary material allowing for the verification of our results.
> > > >
> > > > We struggle to find concrete suggestions in this response, so at the moment we do not know how the reviewer would like us to improve the paper. Regarding the factors mentioned - we do have prompts and contexts in our evaluations (we have reproduced our results on multiple well established benchmarks - CNN/DM, MMLU and Hellaswag), and we do not understand what is practically being meant by testing "inner polysemanticity", "inner feature interactions","external influences", nor how a more "realistic" experiment examining these factors would look like.
> > > >
> > > > We'd greatly appreciate it if the reviewer could back any of the claims with references or a sketch of an experiment we could run so that we can further strengthen our work.

---

> ### Comment · Reviewer_u1oC · 2024-11-29
> **Further concerns wrt the robustness of the findings**
>
> I appreciate the authors' persistency and the code. However, I'd like to reiterate that my concerns are not addressed by the empirical/experiential findings of this submission. I'd like to highlight that my own observations of alternative layer-wise outputs (when probing LLMs with prompts in zero-shot and few-shot settings) also constitute empirical evidence. I ran the script (the logit lens) provided in the supplementary material and obtained the output. Contrary to the claimed "saturated events", the results show that neither the top 1 nor top K tokens maintain their positions across layers. In fact, even the top 1 token fails to hold its position, let alone a consistent picture for ordered top K saturated events (LLM: Meta-Llama-3-8B). This raises concerns about the robustness of the findings.
>
> (Pdb)  text = "What is the location of China's capital?"
>
> (Pdb) logit_ret = logit_lens(tokenizer, model, text, top_k=5)
>
> (Pdb) print_logit_lens(logit_ret)
>
> Layer 1, top_5 logit lens is ['ToPoint', 'ĠKERNEL', 'amac', 'vell', 'Ġà¤ħà¤Ĺ']
>
> Layer 2, top_5 logit lens is ['ĠPitt', 'Ġchin', 'Ġperms', 'agli', 'semb']
>
> Layer 3, top_5 logit lens is ['Ġperms', 'ĠCarson', 'âĢĮØ§ÙĨ', 'BOOLE', 'Ġciv']
>
> Layer 4, top_5 logit lens is ['ĠPitt', 'Ġperms', 'pun', 'ONGL', 'olin']
>
> Layer 5, top_5 logit lens is ['Ġchin', 'NECT', 'ĶåĽŀ', 'Ġredistribute', 'Äįet']
>
> Layer 6, top_5 logit lens is ['#ab', '#ac', 'Sharper', 'TRGL', 'à¹Ħà¸¥à¸Ļ']
>
> Layer 7, top_5 logit lens is ["'gc", 'akah', 'Äįet', 'Enlarge', 'Ġcue']
>
> Layer 8, top_5 logit lens is ['#ab', '#ac', 'Ġdisp', "'gc", 'RetVal']
>
> Layer 9, top_5 logit lens is ['#ab', 'RetVal', "'gc", 'Î»Î¹Îº', 'eph']
>
> Layer 10, top_5 logit lens is ['eph', '#ab', '\\model', '675', 'Ĥ¨']
>
> Layer 11, top_5 logit lens is ['ÑģÑĤÐµ', '485', '#ab', 'Ã³st', 'habit']
>
> Layer 12, top_5 logit lens is ['#ab', '.Xaml', 'taboola', 'VISIBLE', 'RIPT']
>
> Layer 13, top_5 logit lens is ['#ab', 'STYPE', '#ac', '#ad', 'ystate']
>
> Layer 14, top_5 logit lens is ['STYPE', 'HCI', '#ab', '#ad', '#ac']
>
> Layer 15, top_5 logit lens is ['#ad', '#ac', '#ab', 'modifiable', 'STYPE']
>
> Layer 16, top_5 logit lens is ['Ġanswer', 'Î»Î¹Îº', 'ĶåĽŀ', 'ĠHint', '#ab']
>
> Layer 17, top_5 logit lens is ['Ġanswer', 'ĠAnswer', 'ĠHint', 'ModifiedDate', 'weit']
>
> Layer 18, top_5 logit lens is ['Ġanswer', 'ĠAnswer', 'answer', 'ĠANSW', 'Ġanswers']
>
> Layer 19, top_5 logit lens is ['Ġanswer', 'answer', 'ĠAnswer', 'Ġanswered', 'ĠANSW']
>
> Layer 20, top_5 logit lens is ['Ġanswer', 'ĠAnswer', 'answer', 'Ġanswered', 'ĠANSW']
>
> Layer 21, top_5 logit lens is ['Ġanswer', 'ĠAnswer', 'çŃĶæ¡Ī', 'answer', 'Ġanswered']
>
> Layer 22, top_5 logit lens is ['Ġanswer', 'ĠAnswer', 'çŃĶæ¡Ī', 'answer', 'Ġanswered']
>
> Layer 23, top_5 logit lens is ['Ġanswer', 'answer', 'ĠAnswer', 'çŃĶæ¡Ī', 'ĠWhich']
>
> Layer 24, top_5 logit lens is ['Ġanswer', 'Ġwhere', 'Ġwhat', 'ĠWhere', 'Ġwhy']
>
> Layer 25, top_5 logit lens is ['ĠBeijing', 'Ġanswer', 'Ġwhere', 'ĠWhere', 'Ġwhat']
>
> Layer 26, top_5 logit lens is ['ĠBeijing', 'Ġwhere', 'Ġwhat', 'ĠWhere', 'ĠChina']
>
> Layer 27, top_5 logit lens is ['ĠBeijing', 'Ġwhere', 'Ġwhat', 'ĠWhere', 'Ġhow']
>
> Layer 28, top_5 logit lens is ['ĠWhere', 'Ġwhere', 'ĠBeijing', 'ĠWhich', 'Ġwhat']
>
> Layer 29, top_5 logit lens is ['Ġwhere', 'Ġwhat', 'ĠBeijing', 'ĠWhere', 'where']
>
> Layer 30, top_5 logit lens is ['ĠA', 'Ġwhere', 'ĠWhere', 'ĠBeijing', 'ÂłA']
>
> Layer 31, top_5 logit lens is ['ĠWhere', 'ĠA', 'ĠBeijing', 'Ġwhere', 'ĠWhat']
>
> Layer 32, top_5 logit lens is ['ĠBeijing', 'ĠWhat', 'ĠA', 'ĠThe', 'ĠWhere']
>
>
> To verify the "ordered state saturation" phenomenon under more real-world conditions, I propose the following experiments:
>
> 1. In zero-shot settings, use paraphrased prompts (e.g., 5-fold for each prompt) to record changes in top N outputs, simulating real-world prompt variability.
> 2. In few-shot settings, experiment with appended or preceded contexts (including negative examples acting as interference) to examine the impact of in-context learning on the phenomenon.
> 3. When projecting hidden states into the vocabulary space, consider the potential suppression of meaning from other features (BTW "Polysemanticity" is a well-accepted concept in the community of mechanistic interpretability). To better understand the inner mechanism, I suggest exploring SAE-based analytic approaches to uncover potential footprints of "ordered state saturation" in neuron or feature circuits.

---

> > ### Comment · Reviewer_u1oC · 2024-11-29
> > **Further outputs**
> >
> > (Pdb) text = "Can you tell me where the capital of China is?"
> >
> > (Pdb) logit_ret = logit_lens(tokenizer, model, text, top_k=5)
> >
> > (Pdb) print_logit_lens(logit_ret)
> >
> > Layer 1, top_5 logit lens is ['conc', 'amac', 'ĠKERNEL', 'èĦĳ', 'izr']
> >
> > Layer 2, top_5 logit lens is ['YPRE', 'Ġchin', 'Ð³Ð°Ð»Ñĸ', 'ATAB', '#ab']
> >
> > Layer 3, top_5 logit lens is ['#ad', 'Ġchin', 'NECT', "'gc", 'upon']
> >
> > Layer 4, top_5 logit lens is ["'gc", 'ÙĬÙĨØ§', 'HAV', 'Ð³Ð°Ð»Ñĸ', '#ad']
> >
> > Layer 5, top_5 logit lens is ['#ad', 'Ġchin', 'Ġdisag', '#ab', 'NECT']
> >
> > Layer 6, top_5 logit lens is ['TRS', 'gesi', '#ab', 'Ġpent', '#ad']
> >
> > Layer 7, top_5 logit lens is ['.Xaml', 'ĠHaley', 'emain', '_builtin', 'EPROM']
> >
> > Layer 8, top_5 logit lens is ['#ac', 'TRIES', '_builtin', 'EATURE', 'EPROM']
> >
> > Layer 9, top_5 logit lens is ['#ac', '_capabilities', '#ab', 'emain', '_builtin']
> >
> > Layer 10, top_5 logit lens is ['eph', 'è±¡', ');$', 'aje', 'apes']
> >
> > Layer 11, top_5 logit lens is ['Å¯st', ');$', 'agraph', 'KNOWN', 'UFFER']
> >
> > Layer 12, top_5 logit lens is ['#ac', ');$', 'Å¯st', 'VRT', 'esinden']
> >
> > Layer 13, top_5 logit lens is ['_dispatcher', '#ac', 'esinden', 'ãĥ«ãĤ¯', '.ServiceModel']
> >
> > Layer 14, top_5 logit lens is ['Ġbare', 'addon', 'ahren', 'Ã¡zev', 'bare']
> >
> > Layer 15, top_5 logit lens is ['modifiable', 'Ã¡zev', '#ac', 'ç¿°', 'å¼ĺ']
> >
> > Layer 16, top_5 logit lens is ['Ġanswer', 'iem', 'ĠAnswer', 'Ġcorrect', 'ĠBuchanan']
> >
> > Layer 17, top_5 logit lens is ['Ġanswer', 'ĠAnswer', 'ĠdestinationViewController', 'ç¿°', '#ac']
> >
> > Layer 18, top_5 logit lens is ['Ġanswer', 'ĠAnswer', 'ĠdestinationViewController', 'çŃĶæ¡Ī', 'answer']
> >
> > Layer 19, top_5 logit lens is ['Ġanswer', 'ĠAnswer', 'çŃĶæ¡Ī', 'answer', 'ABCDEFGHIJKLMNOP']
> >
> > Layer 20, top_5 logit lens is ['Ġanswer', 'ĠAnswer', 'answer', 'çŃĶæ¡Ī', 'Ġanswered']
> >
> > Layer 21, top_5 logit lens is ['Ġanswer', 'ĠAnswer', 'çŃĶæ¡Ī', 'ĠdestinationViewController', 'ĠHint']
> >
> > Layer 22, top_5 logit lens is ['Ġanswer', 'Ġyes', 'ĠWhere', 'Ġwhere', 'ĠAnswer']
> >
> > Layer 23, top_5 logit lens is ['Ġanswer', 'Ġyes', 'ĠWhich', 'Ġwhere', 'ĠAnswer']
> >
> > Layer 24, top_5 logit lens is ['Ġyes', 'Ġanswer', 'Ġwhere', 'ĠWhere', 'Ã¡zev']
> >
> > Layer 25, top_5 logit lens is ['ĠBeijing', 'ĠChina', 'Ġyes', 'ĠCapitals', 'Ġanswer']
> >
> > Layer 26, top_5 logit lens is ['ĠBeijing', 'Ġcapital', 'ĠCapitals', 'ĠCapital', 'Ġyes']
> >
> > Layer 27, top_5 logit lens is ['ĠBeijing', 'ĠCapitals', 'Ġcapital', 'ĠCapital', 'Ġyes']
> >
> > Layer 28, top_5 logit lens is ['ĠBeijing', 'ĠCapitals', 'Ġyes', 'Ġcapital', 'ĠCapital']
> >
> > Layer 29, top_5 logit lens is ['ĠBeijing', 'Ġyes', 'Ġwhat', 'ĠCapitals', 'Ġcapital']
> >
> > Layer 30, top_5 logit lens is ['ĠBeijing', 'ĠWhere', 'Ġwhere', 'ĠBÄĽ', 'ĠCapitals']
> >
> > Layer 31, top_5 logit lens is ['ĠWhere', 'ĠBeijing', 'ĠWhat', 'ĠIt', 'ĠCan']
> >
> > Layer 32, top_5 logit lens is ['ĠBeijing', 'ĠWhat', 'ĠThe', 'ĠIt', 'ĠI']

---

> ### Author Response · Authors · 2024-11-29
>
> We deeply appreciate the reviewer's willingness to run our code and engage in further discussion. We apologize if we’ve made it seem like saturation events occur for all tokens in all prompts, and acknowledge that although they are prevalent in the data it may not be evident without analyzing a lot of examples.
>
> We note that the reviewer's output was not produced by our code provided in the supplementary materials. We tried our best to remain as true as possible to the reviewer’s points (see code in next comment) and ran the same exact prompts.
>
> We find that in the prompt “What is the location of China's capital?” the following token position predictions reach top-1 saturation at least 3 layers before output:
> * Position 4 (prediction for next-word after ‘location’)  - top-1 saturates at layer 24
> * Position 6 (prediction for next-word after ‘China’) - top-1 saturates at layer 26.
>
> And in the prompt “Can you tell me where the capital of China is?” the following token position predictions reach top-1 saturation at least 3 layers before output:
> * Position 3 (prediction for next-word after ‘tell’) - top-1 saturates at layer 16
> * Position 9 (prediction for next-word after ‘China’) - top-1 saturates at layer 27
> * Position 11 (prediction for next-word after ‘?’) - top-1 saturates at layer 25
>
> At position 3 we can even see some evidence for ordered saturation as the 2nd ranking token reaches saturation at layer 28, and the 4th token reaches saturation at layer 31 (3d token doesn’t undergo saturation).
> Considering that both prompts are very short (10 and 11 tokens, respectively), this is a significant percentage of tokens that undergo top-1 saturation.
>
> Using the same prompts as the reviewer we got different results to their output, which clearly demonstrate the existence of saturation events (at the very least for the top-1 token). The disparity might be due to 3 factors:
> 1. Quantization - as we noted in the appendix and can be seen in our code we use 8-bit quantized version of Llama3 with the following parameters:
> BitsAndBytesConfig(load_in_8bit=True, bnb_4bit_compute_dtype=torch.bfloat16)
> 2. LayerNorm - as is the standard when using logit lens, we apply layer norm to the outputs of each hidden layer before projecting onto the vocabulary space. This is done to ensure consistency with the last layer.
> 3. We look at the model predictions for all token positions in the prompt, and not just the last one.
>
> We reiterate that we do not claim that all predictions undergo top-k saturation, just that enough undergo top-1 saturation for this to be a relevant phenomenon worth exploring.
> As can be seen by running our code on 100 questions randomly sampled from the Hellaswag benchmark, almost 47% of predictions reach top-1 saturation at least 3 layers before output.
> When looking at the next highest ranking tokens, although the numbers get lower they are still a significant fraction of the data (Over 11% of samples for the 2nd highest ranking token, almost 4% of samples for the 3d ranking token, and around 2% of samples for the 4th ranking token).
>
> Regarding the experiments proposed by the reviewer:
>
> 1. To simulate real-world prompt variability we followed the reviewer’s suggestion by:
>     * randomly sampling 200 questions from MMLU dataset,
>     * automatically creating 5 paraphrases for each question with gpt3.5-turbo OpenAI model using the following prompt (resulting in 1K texts): "Paraphrase the following question into 5 different versions:\nQuestion: <Question>\nParaphrases:"
>     * re-running our analysis to validate the existence of saturation events.
>
>     We find that the overall saturation statistics remain almost the same with over 46% of samples reaching saturation at least 3 layers before the output for the top-1 token.
> 2. Can the reviewer explain what is meant here by negative examples? Additionally, as we’ve run our experiments on multiple text datasets (CNN/DM, MMLU and Hellaswag) we would argue that we’ve checked many different contexts, and even shown this phenomenon is **content independent** by reproducing it in a randomly initialized Transformer (for which any context is meaningless) and Transformers from other domains (vision and audio).
> 3. We first note that investigating the inner mechanism that produces the ordered saturation phenomenon implies its existence. In our work, we focus on establishing the generality of this newly discovered phenomenon and propose a high level task-transition mechanism for it, which we validate using both probing and intervention procedures, and even show how it can be leveraged for practical applications. We agree that given the establishment of our claims, a great direction for future work to explore is which feature circuits and specific neurons give rise to this phenomenon, but this is beyond the scope of our paper. We will expand on this topic in the discussion.

---

> > ### Author Response · Authors · 2024-11-29
> >
> > We present our implementation for the functions called in the reviewer’s output to enable further evaluation, and we would appreciate any feedback regarding potential errors or issues in the code.
> >
> > Note: In the line “top_k_indices = torch.topk(layer_logits[-1,:], top_k, dim=-1).indices” the first index can be changed to get the top-k rated predictions at other positions in the prompt.
> >
> > ```python
> > from transformers import AutoTokenizer, AutoModelForCausalLM
> > import torch
> >
> > def tensor_ids_to_tokens(ids):
> >   ids = ids if isinstance(ids, list) else list(ids.squeeze())
> >   tokens = tokenizer.batch_decode(ids)
> >   tokens = [t.strip() for t in tokens]
> >   return tokens
> >
> >
> > def logit_lens(tokenizer, model, text, top_k=5):
> >     inputs = tokenizer(text, return_tensors="pt")
> >     outputs = model(**inputs, output_hidden_states=True)
> >
> >     # Extract hidden states and final logits
> >     hidden_states = outputs.hidden_states
> >     logits = outputs.logits
> >
> >     token_ids = inputs['input_ids'][0]
> >     tokens = tokenizer.convert_ids_to_tokens(token_ids)
> >
> >     # Calculate top-k tokens for each layer
> >     top_k_predictions = {}
> >     for i, hidden_state in enumerate(hidden_states[1:]):
> >         if i != len(hidden_states) -1:
> >           hidden_state = model.model.norm(hidden_state)
> >         layer_logits = model.lm_head(hidden_state)  # Transform hidden states to logits
> >         layer_logits = layer_logits.squeeze()
> >         top_k_indices = torch.topk(layer_logits[-1,:], top_k, dim=-1).indices
> >         top_k_predictions[f"Layer {i+1}"] = tensor_ids_to_tokens(top_k_indices)
> >     return tokens, top_k_predictions
> >
> >
> > def print_logit_lens(logit_ret):
> >     for layer, top_tokens in logit_ret.items():
> >         print(f"{layer}, top_5 logit lens is {top_tokens}")
> >
> >
> > tokens, top_k_predictions = logit_lens(tokenizer, model, "What is the location of China's capital?", top_k=5)
> > print_logit_lens(top_k_predictions)
> > ```

---

### Official Review · Reviewer_iRoJ · 2024-11-03

**Soundness:** 3
**Presentation:** 3
**Contribution:** 3
**Rating:** 8
**Confidence:** 4

**Summary:**

1) This study extends the concept of saturation from only the top-1 token to multiple top-ranked tokens, providing deeper insights into the computational processes within Transformer models.
2) Experiments across diverse domains, including text, vision, and speech, demonstrate the broad applicability of this sequential saturation mechanism.
3) Beyond observing this phenomenon, the authors propose an early-exit algorithm based on their findings, presenting a practical approach to enhance computational efficiency in Transformer models.

**Strengths:**

The paper is written in a clear and understandable manner, with a well-defined approach and valuable findings that go beyond observation to demonstrate practical benefits. I believe this is an good research, and with a few additional points, it would be well-suited for acceptance at the conference.

**Weaknesses:**

If the paper claims that these findings are unique to Transformers (as the title suggests), it should demonstrate that (1) this phenomenon does not occur in other architectures and (2) it consistently appears in recent, state-of-the-art models. Providing this evidence would make the paper much more logically sound and robust.

The practical applicability of the experiments in Sections 5.1 and 5.2 feels somewhat limited; further validation could enhance their real-world relevance. (For example, as shown in A.5, softmax and state cannot outperform ours even at the minimum speedup, which makes the comparison less compelling.)

**Questions:**

(1) Table 1 shows different chance levels for each model (e.g., 25% for Whisper-large vs. 20% for GPT2-XL and ViT-L/16) Could the authors clarify the reason for these differences?

(2) In Figure 3, how did the authors handle cases where two top-ranked tokens reach saturation at the same layer?

(3) Following up, how were such cases handled during classifier training?

(4) In Table 1, how was accuracy calculated in these scenarios?

(5) typo: 400lines: l1(s1) < l2(s2) => l1(s1) < l1(s2)

(6) Could the authors provide a comparison with other early exit methods?

---

> ### Author Response · Authors · 2024-11-15
>
> We're glad you found our paper clear and understandable, with a well-defined approach and practical contributions.
>
> We do not claim that the ordered saturation phenomenon is unique to Transformers, which, as we mention in the “Conclusion and Future Work” section, is a question worth exploring, particularly in RNNs. We argue that it is inherent to the architecture and support our claim by reproducing the findings in an untrained randomly initialized Transformer model (Section 4.1).
>
> Following your comments and to demonstrate that this phenomenon appears in recent, state-of-the-art models, we have reproduced our results using Llama3-8B on MMLU and Hellaswag benchmarks. We find that Llama3’s top-k tokens reach saturation in order of their ranking up to (and including) the fourth-ranking token and that we’re able to predict the task number from layer embeddings with 88% accuracy. Please see Figure 7 and Table 5 in Section B.2 of the appendix for more details.
>
> You noted that the practical applications require further validation to enhance their real-world relevance. Could you please explain what type of validation would be helpful here?
>
> Regarding your questions:
> 1. The difference in chance level for each model follows from the different number of tasks used when training the probing classifier for each model. The number of tasks is determined per model to be the maximum number for which, after balancing the data, there are at least 10 embeddings from each layer in each class from at least 4 different layers. To clarify this point, we’ve added the relevant footnote to Section 2.3.
>
> 2. In cases where two top-ranked tokens reached saturation as the same layer, they both get the same ranking which works against our claim of ordered saturation. Despite this, we see a statistically significant difference between the average ranks of adjacent top tokens.
>
> 3. When creating the data for the task number classifier, if two adjacent top tokens reached saturation at the same layer (for example 3d and 4th) then these would be labeled as belonging to the task of the smaller token between them (in this case 3d task). This adds some noise to the data, but does not prevent the classifier from achieving high accuracy.
>
> 4. Classifier accuracy was calculated as the percentage of samples where the predicted label matches exactly that of the true label.
>
> 5. Fixed typo.
>
> 6. We provided comparison with the only known token level early exit methods known to us based on the cited survey [1] . If you can suggest others, we would be happy to compare our strategy against them as well.
>
> We will add clarifications to all of these questions in the paper.
>
> If there are specific models, benchmarks, or analyses you would like us to explore to strengthen the work further, please do not hesitate to let us know.
>
> [1] - A survey on efficient inference for large language models. (https://arxiv.org/abs/2404.14294)

---

> > ### Author Response · Authors · 2024-11-20
> >
> > Dear Reviewer,
> >
> > Thank you for your feedback on our submission. We have carefully addressed all your comments and provided detailed responses to each of your concerns. In addition, we’ve reproduced our ordered saturation, task transition probing and task switching intervention results on  ***Mistral*** and  ***Falcon*** models (see Section B.4 in the Appendix) to further show the generality of our findings. We hope that our replies clarify the points raised and strengthen the case for our paper.
> >
> > As the discussion period concludes on ***November 26***, we would greatly appreciate any further feedback, comments, or follow-up questions you may have. Your input is invaluable in helping us improve our work.
> >
> > Thank you once again for your time and effort.

---

> ### Comment · Reviewer_iRoJ · 2024-11-27
>
> Thank you to the authors for engaging in discussion. I appreciate the time that put in to address my concerns and feel they were sufficiently addressed, so I raised my score.

---

### Official Review · Reviewer_8PE6 · 2024-11-03

**Soundness:** 4
**Presentation:** 3
**Contribution:** 4
**Rating:** 8
**Confidence:** 4

**Summary:**

This paper investigates the internal mechanisms of Transformers, particularly focusing on the computation performed after the top-1 prediction has been determined, a phenomenon referred to as the "saturation event." The authors expand the concept of saturation events to top-k tokens and demonstrate that these events occur in a ranked order across various modalities (language, vision, speech) and Transformer architectures. They propose a task-transition mechanism to explain this sequential saturation, where each task corresponds to predicting the k-th most probable token. The paper further shows that it is possible to predict the current task from hidden layer embeddings and to induce task transitions through interventions. Finally, the authors introduce a token-level early-exit strategy that improves performance and efficiency, outperforming existing methods.

**Strengths:**

1. The paper provides a fresh perspective on how Transformers process predictions beyond the top-1 token, which is a significant contribution to the field of model interpretability.
2. The study's scope extends across multiple modalities, enhancing the generalizability of the findings and demonstrating the robustness of the proposed mechanisms.
3. The authors back their claims with extensive experiments and provide empirical evidence supporting the task-transition hypothesis.
4. The paper not only has theoretic analysis but also explores practical applications, such as the early-exit strategy, which has potential implications for improving model efficiency and accuracy.

**Weaknesses:**

This is a suggestion rather than a weakness.

It would be great to discuss whether this is a fundamental solution for early exit problem. After all, it is ridiculous for us to let problems with all level of difficulties to go through all the layers. There must be a solution that significantly improve the inference efficiency of the current LLM through sparsity or early exit. Yet the solution proposed by this paper give only marginal improvement.

**Questions:**

Please see the weakness part. Highly appreciate if the author give some feedback.

---

> ### Author Response · Authors · 2024-11-15
>
> We are pleased that you appreciated our contributions to advancing the understanding of Transformer models. It is encouraging to see that our efforts to establish generalizability across modalities and provide empirical backing for the task-transition hypothesis were well received.
>
> Although we provide a new early-exit strategy as a practical application for our findings, this is not the main focus of our paper. Furthermore, our proposed strategy focuses only on the top-1 as is the standard in most efficiency focused works, but as we argue in the Related Works section, this does not take into account the other top-k tokens which are often used in common generation sampling methods. For this reason, to fundamentally improve inference efficiency as you suggest, future work should consider the trade-off between speedup and top-k accuracy.
>
> If there are specific models, benchmarks, or analyses you would like us to explore to strengthen the work further, please do not hesitate to let us know

---

> > ### Author Response · Authors · 2024-11-20
> >
> > Dear Reviewer,
> >
> > Thank you for your feedback on our submission. We have carefully addressed all your comments and provided detailed responses to each of your concerns. In addition, we’ve reproduced our ordered saturation, task transition probing and task switching intervention results on  ***Mistral*** and  ***Falcon*** models (see Section B.4 in the Appendix) to further show the generality of our findings. We hope that our replies clarify the points raised and strengthen the case for our paper.
> >
> > As the discussion period concludes on ***November 26***, we would greatly appreciate any further feedback, comments, or follow-up questions you may have. Your input is invaluable in helping us improve our work.
> >
> > Thank you once again for your time and effort.

---

> > ### Comment · Reviewer_8PE6 · 2024-11-22
> >
> > Can you elaborate on this sentence "should consider the trade-off between speedup and top-k accuracy."? It would be great if you can discuss some related work about this.

---

> ### Author Response · Authors · 2024-11-23
>
> Thank you for the opportunity to elaborate on this point. When we mention the trade-off between speedup and top-k accuracy, we are emphasizing the importance of considering how early-exit strategies affect more than just top-1 predictions. As discussed in Section 6 of our paper, although greedy decoding (which uses top-1 predictions) is considered suboptimal for many language generation tasks compared to stochastic sampling [1], it remains the standard approach in early-exit evaluations, including in recent works such as [2]. Even early-exit papers that use top-k or top-p sampling techniques during evaluation such as [3,4] treat them as a technical detail. Based on our work it is possible to establish which top-k tokens reach saturation and could be relevant predictions. Future research should explore these sampling strategies more thoroughly, considering them as key hyperparameters to strike a balance between efficiency and maintaining the quality of generated outputs.
>
> We appreciate your thoughtful question, as it underscores an important implication of our work. We will incorporate this discussion into the conclusion of our paper to emphasize its relevance to the field.
>
> [1] A Thorough Examination of Decoding Methods in the Era of LLMs - https://arxiv.org/abs/2402.06925
>
> [2] LayerSkip: Enabling Early Exit Inference and Self-Speculative Decoding - https://arxiv.org/pdf/2404.16710
>
> [3] SkipDecode: Autoregressive Skip Decoding with Batching and Caching for Efficient LLM Inference -  https://arxiv.org/pdf/2307.02628
>
> [4] ConsistentEE: A Consistent and Hardness-Guided Early Exiting Method for
> Accelerating Language Models Inference - https://ojs.aaai.org/index.php/AAAI/article/download/29922/31612

---

### Official Review · Reviewer_w56e · 2024-11-04

**Soundness:** 2
**Presentation:** 3
**Contribution:** 2
**Rating:** 5
**Confidence:** 3

**Summary:**

This paper is to understand the inner workings of transformers to achieve more accurate and efficient predictions. Authors expand the concept of saturation events for top-k tokens, demonstrating that similar saturation events occur across language, vision, and speech models. The experiments show that it is possible to predict the current task from hidden layer embedding. Furthermore, authors use an intervention method to cause the model to swithc from one task to the next.

**Strengths:**

1. This paper is well written with clear illustrations.
2. There are extensive experiments across different modalities, such as vision, language, and speech.
3. Compared with two baseline models, the proposed method demonstrates improved performance.

**Weaknesses:**

1. Compared with Line 275 and L 276, the proposed method seems to be model-specific. While the results in Table 1 show the layer embeddings contain information about the task number, such information may only work for language, but not vision and speech.

2. in Line 472, why the second token's saturation layer is at least "7" layers before the output? The hypothesis seems to be hand-crafted without any supportive evidence.

3. In Table 2, the comparisons of different strategies are a little bit weak to support the claim. Is 40% accuracy a significant improvement compared to the baselines 35.9% and 37.5%? As these numbers are pretty low to have any real applications.

**Questions:**

See the above weakness. The main concern is that the proposed method is quite heuristic and hand-crafted without enough supportive evidence. While the authors demonstrate the results across different modalities, it is difficult to evaluate the effectiveness of this method given the relatively low numbers.

---

> ### Author Response · Authors · 2024-11-15
>
> We are glad you found our writing and illustrations clear and noted the extensive experiments across different modalities.
>
> Answers to points raised:
> 1. Although the classifiers for vision and speech models achieve lower accuracy than that of the classifier trained on the language model, the results are still statistically significant with p < 0.00001 based on a Binomial Distribution probability test. The difference in accuracy might be due to the different number of layers between the models, where GPT2 has 48 layers, ViT and Whisper have 32. The larger number of layers lends itself to more layers per task and better separation between them.
>
> 2. The number 7 was chosen arbitrarily as a hyperparameter; to demonstrate the robustness of the effect, we re-ran the experiment with values in the range 2-10 and indeed observed the same phenomenon, indicating that there’s nothing unique about the number 7: the accuracy of 2nd tokens which reach saturation i layers before output   (with 2 <=i <=10) is significantly higher than of those that are determined only at the last layer. Please see Section B.3 in the Appendix for more details.
>
> 3. Thank you for bringing this to our attention. After running an independent samples t-test, we found that the difference in accuracy between our method and each of the alternatives is statistically significant, with p < 0.001 and made note of it in Table 2.
> We would like to emphasize that the practical applications are not the main focus of our paper, and are meant to showcase how our theoretical insights can be used.
>
> You mentioned that the proposed method seems heuristic and hand-crafted, and that there is not enough supportive evidence. Can you please expand on what you mean by that? Which method used in the paper are you referring to and what additional supportive evidence could we provide?
>
> If there are specific models, benchmarks, or analyses you would like us to explore to strengthen the work further, please do not hesitate to let us know.

---

> > ### Author Response · Authors · 2024-11-20
> >
> > Dear Reviewer,
> >
> > Thank you for your feedback on our submission. We have carefully addressed all your comments and provided detailed responses to each of your concerns. In addition, we’ve reproduced our ordered saturation, task transition probing and task switching intervention results on  ***Mistral*** and  ***Falcon*** models (see Section B.4 in the Appendix) to further show the generality of our findings. We hope that our replies clarify the points raised and strengthen the case for our paper.
> >
> > As the discussion period concludes on ***November 26***, we would greatly appreciate any further feedback, comments, or follow-up questions you may have. Your input is invaluable in helping us improve our work.
> >
> > Thank you once again for your time and effort.

---

> > > ### Author Response · Authors · 2024-12-02
> > >
> > > We noticed there was no engagement from your side during the discussion phase, despite the conference extending this period to encourage dialogue. If there are any lingering questions or points of misunderstanding, we would greatly appreciate the opportunity to address them. Your feedback is crucial for improving our work, and we thank you for your time and consideration.

---

### Official Review · Reviewer_nRZW · 2024-11-04

**Soundness:** 3
**Presentation:** 2
**Contribution:** 4
**Rating:** 8
**Confidence:** 3

**Summary:**

This work expands the top-1 saturation event to top-k tokens and finds a significant phenomenon that there is an order law between the saturation events and the corresponding tokens' ranking. Additionally, such phenomenon is rooted in transformer architecture, which happens across all modality and even in untrained models. With the finding of such order law on the saturation events, an efficient decoding method is proposed for early-exit for next-token prediction.

**Strengths:**

1. The concept and research problem in this work is very novel. It significantly extends the range of saturation event to the top-k token level.

2. The findings are very brute-force, which can be easily used for early-exit decoding for next token prediction acceleration without introducing much additional computation. And such decoding strategy is general and easily applied across all transformer based LLMs.

**Weaknesses:**

1. The evaluation of this method is limited in its size and targeted models. 60k tokens and 100 texts are a too small evaluation set size for a robust conclusion. GPT2-XL is also not well-trained and the experiments on Llama-3 is more recommended. If the saturation event for top-k tokens only happens at very late layers, then the acceleration ratio is not that ideal compared with GPT2-XL experiments. CNN summarization task is a very basic task for recent LLMs, while the benchmarks of MMLU, Hellaswag are more recommended for study and experiments.

**Questions:**

1. I recommend to change the definition of "task" to other words like "process". The concept of "task transition" makes me feel like there is some generality in different tasks like summarization, QA, dialogues, etc.

2. The experiments on vision and speech transformers are great demonstrations to your claim on the generality of your findings but have no contribution to real applications like decoding acceleration. Maybe you can consider to remove them into appendix and enlarge your study on language model part.

3. I am very interested in whether such saturation event happens on all tokens in the context or just a few tokens? If it does not happen on all tokens, how you can deal with the prediction towards these non-saturation tokens?

---

> ### Author Response · Authors · 2024-11-15
>
> We’re glad that you found our concept novel and see the potential of our proposed practical applications.
>
> Per your suggestion we re-ran our order saturation and task transition probing experiments using Llama3-8B model on 1K randomly chosen samples from MMLU and Hellaswag benchmarks. We show that Llama3’s top-k tokens reach saturation in order of their ranking up to (and including) the 4th ranking token, and that we’re able to predict the task number from layer embeddings with 88% accuracy. Please see Section B.2 in the appendix for more details,
> And let us know if you feel that more samples or additional datasets are needed.
>
> Regarding your questions:
> 1. We appreciate your suggestion, and acknowledge that in the NLP field the word “task” has other more downstream connotations. Yet we feel that the word “process” here  would be misleading, and that “task” better encapsulates the discrete nature of the proposed mechanism, where the model determines each token in sequence.
>
> 2. We recognize that our practical applications sections apply only to language models, but we feel that the main focus of our paper is the discovery of the ordered saturation phenomenon and the proposed task transition mechanism, which as we argue are inherent to Transformer architecture regardless of modality. For this reason we think that focusing only on language models would be doing a disservice to the generality of our findings.
>
> 3. To address the question of which tokens undergo saturation we’ve added a section to the Appendix (B.1) where we show  that in GPT2 saturation events are common for top-k tokens, with over 80% of samples reaching top-1 saturation. Additionally, these samples belong to different parts of speech (POS), and are not just function words for example, with over 27% of them being nouns.
> Our findings do not apply to samples which do not reach top-1 saturation, but our results in Section 5.2 suggest that these predictions night be less accurate in terms of language modeling.
>
> If there are specific models, benchmarks, or analyses you would like us to explore to strengthen the work further, please do not hesitate to let us know.

---

> > ### Author Response · Authors · 2024-11-20
> >
> > Dear Reviewer,
> >
> > Thank you for your feedback on our submission. We have carefully addressed all your comments and provided detailed responses to each of your concerns. In addition, we’ve reproduced our ordered saturation, task transition probing and task switching intervention results on  ***Mistral*** and  ***Falcon*** models (see Section B.4 in the Appendix) to further show the generality of our findings. We hope that our replies clarify the points raised and strengthen the case for our paper.
> >
> > As the discussion period concludes on ***November 26***, we would greatly appreciate any further feedback, comments, or follow-up questions you may have. Your input is invaluable in helping us improve our work.
> >
> > Thank you once again for your time and effort.

---

> > > ### Comment · Reviewer_nRZW · 2024-11-28
> > >
> > > Thanks for the additional results on Llama3 and the helpful responses. I raised my score to 8.

---

### Author Response · Authors · 2024-11-15

We thank our reviewers for their insightful feedback and helpful suggestions.

In light of their comments, we’ve run additional experiments, which are presented in Appendix B:

1. To provide evidence establishing how common saturation events are in the data, we’ve added Section B.1 showing that in GPT2 model  over 80% of samples reach top-1 saturation.

2. We have reproduced our results using Llama3-8B on MMLU and Hellaswag benchmarks to demonstrate the ordered saturation phenomenon in recent state-of-the-art models. We find that Llama3’s top-k tokens reach saturation in order of their ranking up to (and including) the fourth-ranking token and that we can predict the task number from layer embeddings with 88% accuracy. Please see Figure 7 and Table 5 in Section B.2 of the appendix for more details.

3. In Section 5.2, the number 7 was chosen randomly as an hyperparameter; to demonstrate the robustness of the effect we re-ran the experiment with values in the range 2-10 and indeed observed the same phenomenon, indicating that there’s nothing unique about the number 7 (see Section B.3).

We’ve also added some clarifications and statistical significance markers in the paper itself, color-coded in cyan for your convenience.

Finally, we uploaded a Jupyter notebook to supplementary materials called “saturation_events.ipynb”, which allows the reviewers to verify the existence and frequency of the saturation events discussed in the paper on Llama3 model.

---

> ### Author Response · Authors · 2024-11-27
>
> In addition, we’ve reproduced our preliminary practical applications results showing that our insights can be leveraged for better efficiency and performance in LLMs using Llama3, Mistral and Falcon models (Sections B.2.1, B.4.1 and B.4.2 respectively in the Appendix) to further show the generality of our findings:
>
> * We show that our novel early-exit strategy based on our task-transition classifier outperforms SOTA softmax-response and hidden-state saturation methods in Llama3 and Mistral, and outperforms softmax-response in Falcon in terms of the efficiency and accuracy tradeoff.
>
> * We show that the saturation of the second highest ranking token affects the accuracy of it’s language modeling (when the top-1 token is incorrect), so that the difference in next-word prediction accuracy between 2nd token achieving saturation i layers before final layer (with 2 <= i <= 6) condition and the condition of the 2nd token being determined only in the last layer is statistically significant with p < 0.001 even when correcting for multiple comparisons.

---

### Author Response · Authors · 2024-12-04

Dear Reviewers,

Thank you for your insightful feedback and thoughtful engagement during the review process.

To further support your evaluation, we have created an anonymous Colab notebook that allows you to verify the ordered saturation phenomenon of top-k tokens discussed in our paper. This notebook enables replication of results on the GPT-2 XL model with the MMLU dataset and is optimized to run on the free version of Colab using a T4 machine.

You can access the notebook here:
https://colab.research.google.com/drive/1qzhLcfeScbkTu99sq0SRWnlgKX9bBQ6h#scrollTo=Dcj0f03mLIiI

We hope this resource enhances your review experience and are grateful for the time and effort you’ve invested in reviewing our work.

---

### Meta-Review · Area_Chair_gcFZ · 2024-12-21

**Metareview:**

This paper receives the mixed ratings of 8, 5, 3, 8, 8. In this paper, the authors introduce a concept of "sequential saturation" in Transformers, analyzing how top-k tokens are fixed layer-by-layer beyond the top-1 prediction. This paper provides evidence supporting that saturation of top-ranking tokens occurs sequentially in various Transformer architectures and modalities. Furthermore, a task-transition mechanism is introduced to explain this phenomenon, validated through interventions and probing experiments.

Strengths:
- The idea of sequential saturation is intriguing, expanding current understanding of Transformer behavior.
- Extensive experiments across text, vision, and speech modalities strengthen the claim of generalizability.
- The proposed applications demonstrate potential practical utility in improving Transformer efficiency.

Area of improvements:
- The concept of task transitions, while innovative, lacks a clear connection to practical modeling scenarios. The assumption that each token represents a discrete task oversimplifies real-world complexities.
- Evidence from untrained models, while theoretically insightful, weakens applicability to real-world, pre-trained settings.
- Reviewer u1oC noted significant discrepancies when attempting to reproduce key results, raising concerns about the robustness of the experiments. The reliance on specific datasets and hyperparameters was not sufficiently detailed in the main paper, making reproducibility challenging.
- While statistical metrics like Kendall’s tau and probing accuracies are provided, the lack of clear visual/conceptual explanations for task transitions weaken the interpretability of the claims.
- Early-exit strategy, demonstrate only incremental gains over baseline methods.

The paper explores an interesting and promising direction, offering an innovative perspective on Transformer dynamics. However, concerns regarding the model's realism and the reproducibility of results persist despite the rebuttal and discussion. We encourage the authors to address these issues comprehensively, as doing so will significantly strengthen the work for a future resubmission.

**Additional Comments On Reviewer Discussion:**

The authors and reviewers engaged in active and productive discussions, which helped clarify several aspects of the submission. The authors provided additional details to address questions regarding the experimental setup and the proposed task-transition mechanism. However, despite these efforts, some concerns raised by the reviewers, particularly around the reproducibility of results and the practical realism of the model, remain unresolved. Reviewer u1oC highlighted significant challenges in reproducing the experimental findings, which were not fully alleviated by the rebuttal.

The discussion brought greater transparency to certain methodological choices, and at the same time, pointed out the areas where further empirical evidence and clarity are necessary. These interactions demonstrate the potential of this work, and we encourage the authors to build on the feedback received to strengthen their future submissions.

---

### Decision · Program_Chairs · 2025-01-22

Reject